# Asynchronous Proportional Response Dynamics: Convergence in Markets with Adversarial Scheduling

**Yoav Kolumbus**
Cornell University
yoav.kolumbus@cornell.edu

**Menahem Levy**
The Hebrew University of Jerusalem
menahem.levy@mail.huji.ac.il

**Noam Nisan**
The Hebrew University of Jerusalem
noam@cs.huji.ac.il

## Abstract

We study Proportional Response Dynamics (PRD) in linear Fisher markets, where participants act asynchronously. We model this scenario as a sequential process in which at each step, an adversary selects a subset of the players to update their bids, subject to liveness constraints. We show that if every bidder individually applies the PRD update rule whenever they are included in the group of bidders selected by the adversary, then, in the generic case, the entire dynamic converges to a competitive equilibrium of the market. Our proof technique reveals additional properties of linear Fisher markets, such as the uniqueness of the market equilibrium for generic parameters and the convergence of associated no swap regret dynamics and best response dynamics under certain conditions.

## 1 Introduction

A central notion in the study of markets is the *equilibrium*: a state of affairs where no single party wishes to unilaterally deviate from it. The main benefit of focusing on the notion of equilibria lies in what it ignores: how the market can reach an equilibrium (if at all). This latter question is obviously of much interest as well, especially when considering computational aspects,[1] and a significant amount of research has been dedicated to the study of "market dynamics" and their possible convergence to an equilibrium. Almost all works that study market dynamics consider synchronous dynamics.

**Synchronous Dynamics:** At each time step $t$, all participants simultaneously update their behavior based on the state at time $t - 1$.

Such synchronization is clearly difficult to achieve in real markets, and so one might naturally wonder to what extent full synchrony is needed or whether convergence of market dynamics occurs even asynchronously. There are various possible levels of generality of asynchrony to consider. The simplest model considers a sequential scenario in which, at every time step $t$, an *adversary* chooses a single participant, and only this participant updates their behavior based on the state at time $t - 1$. The adversary is constrained to adhere to some liveness condition, such as scheduling every participant infinitely often or at least once every $T$ steps. In the most general model [59], the adversary may also introduce message delays, leading players to respond to dated information. In this paper, we focus on an intermediate level of permissible asynchrony where updates may occur in an arbitrary asynchronous manner, but message delays are always shorter than the granularity of activation. In our proofs, the adversary's goal is to select the schedule of strategy updates by the players in a way that prevents the dynamic from converging to a market equilibrium.

---

[1]As we know that finding an equilibrium may be computationally intractable in general.

37th Conference on Neural Information Processing Systems (NeurIPS 2023).

**Activation Asynchrony:**[2] Every time step $t$, an arbitrary subset of participants is chosen by the adversary and all of these participants update their behavior based on the state at time $t - 1$. The adversary must adhere to the liveness condition where for every participant some set that includes them must be chosen at least once every $T$ consecutive steps.

The market dynamics that we study in this paper are linear Fisher markets [9] with proportional response dynamics (PRD), a model that has received much previous attention [5, 10, 19, 74] and for which synchronous convergence to equilibrium is known. While there are a few asynchronous convergence results known for other dynamics, specifically for tatonnement dynamics [18, 23], there are no such results known for proportional response dynamics, and achieving such results has been mentioned as an open problem in [19, 74].

**Fisher Markets with Linear Utilities:** There are $n$ players and $m$ goods. Each player $i$ has a budget $B_i$ and each good $j$ has, w.l.o.g., a total quantity of one. Buyer $i$'s utility from getting an allocation $x_i = (x_{i1}, ..., x_{im})$ is given by $u_i(x_i) = \sum_j a_{ij} x_{ij}$, where the parameters $a_{ij} \geq 0$ are part of the definition of the market. A market equilibrium is an allocation $X = (x_{ij})$ (where $0 \leq x_{ij} \leq 1$) and a pricing $p = (p_j)$ with the following properties. (1) Market clearing: for every good $j$ it holds that $\sum_i x_{ij} = 1$; (2) budget feasibility: for every player $i$ it holds that $\sum_j x_{ij} p_j \leq B_i$; and (3) utility maximization: for every player $i$ and every alternative allocation $y = (y_1, ..., y_m)$ with $\sum_j y_j p_j \leq B_i$ we have that $u_i(x_i) \geq u_i(y)$.

**Proportional Response Dynamics:** At each time step $t$, each player $i$ will make a bid $b_{ij}^t \geq 0$ for every good $j$, where $\sum_j b_{ij}^t = B_i$. In the first step, the bid is arbitrary. Once bids for time $t$ are announced, we calculate $p_j^t = \sum_i b_{ij}^t$ and allocate the goods proportionally: $x_{ij}^t = b_{ij}^t / p_j^t$, providing each player $i$ with utility $u_i^t = \sum_j a_{ij} x_{ij}^t$. At this point, player $i$ updates its bids for the next step by bidding on each good proportionally to the utility obtained from that good: $b_{ij}^{t+1} = B_i \cdot a_{ij} x_{ij}^t / u_i^t$.

From the perspective of the player, proportional response updates can be thought of as a simple parameter-free online learning heuristic, with some similarity to regret-matching [43] in its proportional updates, but one that considers the utilities directly, rather than the more sophisticated regret vector loss.

It is not difficult to see that a fixed point of this proportional response dynamic is indeed an equilibrium of the Fisher market. Significantly, it was shown in [5] that this dynamic does converge, in the synchronous model, to an equilibrium. As mentioned, the question of asynchronous convergence was left open. We provide the first analysis of proportional response dynamics in the asynchronous setting, and provide a positive answer to this open question in our "intermdiate" level of asynchrony.

**Theorem 1.** *For linear Fisher markets, proportional response dynamics with adversarial activation asynchrony, where each player is activated at least once every $T$ steps, approach the set of market equilibrium bid profiles. The prices in the dynamics converge to the unique equilibrium prices.*

The dynamics approaching the set of equilibria means that the distance between the bids at time $t$ and the set of market equilibrium bid profiles converges to zero as $t \to \infty$. Additionally, in Section 7, we show that for generic parameters (i.e., except for measure zero of possible $(a_{ij})$'s), the market equilibrium bid profile is unique, and thus Theorem 1 implies convergence of the bids to a point in the strategy space. We do not know whether the genericity condition is required for asynchronous convergence of the bids to a point, and we leave this as a minor open problem. We did not analyze the rate of convergence to equilibrium; we leave such analysis as a second open problem. Our main open problem, however, is the generalization to full asynchrony with message delays.

**Open Problem:** Does such convergence occur also in the full asynchronous model where the adversary may introduce arbitrary message delays?

Our techniques rely on considering an associated game obtained by using "modified" utility functions for each of the players: $\tilde{u}_i(b) = \sum_j b_{ij} \ln(a_{ij}) + \sum_j p_j(1 - \ln(p_j))$. We show that a competitive market equilibrium (with the original utility functions) corresponds to a Nash equilibrium in the associated game.[3] These modified utility functions are an adaptation to an individual utility of a

---

[2] In [59] this was termed "simultaneous."

[3] It is worthwhile to emphasize, though, that a competitive market equilibrium is not a Nash equilibrium in the original market, since the players are price takers rather than fully rational. See Section 3.

function $\Phi(b) = \sum_{ij} b_{ij} \ln(a_{ij}) + \sum_j p_j(1 - \ln(p_j))$ that was proposed in [65] as an objective for a convex program for equilibrium computation.[4] This function was first linked with proportional response dynamics in [5] where it was proven that synchronous proportional response dynamics act as mirror descent on this function. We show that $\Phi$ serves as a potential in our associated game.

**Theorem 2.** *The following three sets of bid profiles are identical: (1) the set of pure strategy Nash equilibria of the associated game, (2) the set of market equilibria of the Fisher market, and (3) the maximizing set of the potential function $\Phi$.*

The technical core of our proof is to show that not only does a synchronized proportional response step by *all* the players increase the potential function $\Phi$ but, in fact, every proportional response step by any *subset of the players* increases this potential function.

The point of view of market equilibria as Nash equilibria of the associated game offers several other advantages, e.g., suggesting several other dynamics that are related to proportional response that can be of interest. For example, we show that letting players best-respond in the associated game corresponds to the limit of a sequence of proportional response steps by a single player, but can be implemented as a single step of such a best-response, which can be computed efficiently by the players and may converge faster to the market equilibrium. Another possibility is using some (internal) regret-minimization dynamics (for the associated game), which would also converge to equilibrium in the generic case since, applying [56], it is the unique correlated Equilibrium as well.

The structure of the rest of the paper is as follows. In Section 2, we provide further formal details and notations that will be useful for our analysis. In Section 3, we present the associated game and its relation to the competitive equilibria of the market. In Section 4, we study best response dynamics in the associated game and their relation to PRD. In Section 5, we present a key lemma regarding the potential function of the associated game under bid updates by subsets of the players. Then, in Section 6 we complete our proof of convergence for asynchronous PRD. In Section 7, we show the uniqueness of the market equilibrium for generic markets. In Section 8, we provide simulation results that compare the convergence of proportional response dynamics with best response dynamics in the associated game in terms of the actual economic parameters in the market, namely, social welfare and the convergence of the bid profiles. Finally, in Section 9, we conclude and discuss the limitations of our technique and open questions. All proofs in this paper are deferred to the appendix.

### 1.1 Further Related Work

Proportional response dynamics (PRD) were originally studied in the context of bandwidth allocation in file-sharing systems, where it was proven to converge to equilibrium, albeit only for a restrictive setting [71]. Since then, PRD has been studied in a variety of other contexts, including Fisher markets, linear exchange economies, and production markets. See [10] for further references. In Fisher markets, synchronous PRD has been shown to converge to market equilibrium for Constant Elasticity of Substitution (CES) utilities in the substitutes regime [74]. For the linear Fisher market setting, synchronous PRD was explained in [5] as mirror descent on a convex program, previously discovered while developing an algorithm to compute the market equilibrium [65], and later proven to be equivalent to the famous Eisenberg-Gale program [22]. By advancing the approach of citebirnbaum2011distributed, synchronous PRD with mild modifications was proven to converge to a market equilibrium for CES utilities in the complements regime as well [19]. In linear exchange economies, synchronous PRD has been shown to converge to equilibrium in the space of utilities and allocations while prices do not converge and may cycle, whereas for a damped version of PRD, also the prices converge [11]. In production markets, synchronous PRD has been shown to increase both growth and inequalities in the market [13]. PRD has also been proven to converge with quasi-linear utilities [38], and to remain close to market equilibrium for markets with parameters that vary over time [20]. All the above works consider simultaneous updates by all the players, and the question of analyzing asynchronous dynamics and whether they converge was raised by several authors as an open problem [19, 74].

Asynchronous dynamics in markets have been the subject of study in several recent works. However, these works examine different models and dynamics than ours, and to our knowledge, our work presents the first analysis of asynchronous proportional response bidding dynamics. In [23], it is

---

[4]Notice that $\Phi$ is not the sum of the $\tilde{u}_i$'s, as the second term appears only once.

shown that tatonnement dynamics under the activation asynchrony model converge to equilibrium, with results for settings both with and without carryover effects between time units. A later work, [18], showed that tatonnement price dynamics converge to a market equilibrium under a model of sequential activation where in every step a single agent is activated, and where additionally, the information available to the activated seller about the current demand may be inaccurate within some interval that depends on the range of past demands. A different approach taken in [29] assumes that each seller has a set of rules affected by the other players' actions and governing its price updates; it is shown that the dynamics in which sellers update the prices based on such rules converge to a unique equilibrium of prices in the activation asynchrony model.

Online learning with delayed feedback has been studied for a single agent in [61], which established regret bounds that are sub-linear in the total delay. A different approach, presented in [52], extends online gradient descent by estimating gradients when the reward of an action is not yet available due to delays. When considering multiple agents, [75] extended the results of [61]. They demonstrated that in continuous games with particular stability property, if agents experience equal delays in each round, online mirror descent converges to the set of Nash equilibria and a modification of the algorithm converges even when delays are not equal.

Classic results regarding the computation of competitive equilibria in markets mostly consider centralized computation and vary from combinatorial approaches using flow networks [26–28, 45], interior point [72], and ellipsoid [21, 44] methods, and many more [3, 25, 26, 30, 39, 40, 63]. Eisenberg and Gale devised a convex program which captures competitive equilibria of the Fisher model as its solution [31]. Notable also is the tatonnement model of price convergence in markets dated back to Walras [70] and studied extensively from Arrow [1] and in later works.

More broadly, in the game theoretic literature, our study is related to a long line of work on learning in games, starting from seminal works in the 1950s [6, 14, 42, 62], and continuing to be an active field of theoretical research [24, 35, 49, 60, 66], also covering a wide range of classic economic settings including competition in markets [7, 54], bilateral trade [16, 32], and auctions [2, 34, 48], as well as applications such as blockchain fee markets [50, 51, 57] and strategic queuing systems [36, 37]. For a broad introduction to the field of learning in games, see [17, 43]. The vast majority of this literature studies repeated games under the synchronous dynamics model. Notable examples of analyses of games with asynchronous dynamics are [46], which study best response dynamics with sequential activation, and [59], which explore best response dynamics in a full asynchrony setting which includes also information delays, and show that in a class of games called max-solvable, convergence of best response dynamics is guaranteed. Our analysis of best response dynamics in Section 4 takes a different route, and does not conclude whether the associated game that we study is max-solvable or not; such an analysis seems to require new ideas.

Our work is also related to a large literature on asynchronous distributed algorithms. We refer to a survey on this literature [41]. The liveness constraint that we consider in the dynamics[5] is related to those, e.g., in [4, 67]. Recent works that are conceptually more closely related are [15, 69, 73], which propose asynchronous distributed algorithms for computing Nash equilibria in network games. Notably, [15] propose an algorithm that converges to an equilibrium in a large class of games in asynchronous settings with information delays. Their approach, however, does not capture proportional response dynamics and does not apply to our case of linear Fisher markets.

## 2   Model and Preliminaries

**The Fisher market**: We consider the classic Fisher model of a networked market in which there is a set of buyers $\mathcal{B}$ and a set of divisible goods $\mathcal{G}$. We denote the number of buyers and number of goods as $n = |\mathcal{B}|$, $m = |\mathcal{G}|$, respectively, and index buyers with $i$ and goods with $j$. Buyers are assigned budgets $B_i \in \mathbb{R}^+$ and have some value[6] $a_{ij} \geq 0$ for each good $j$. Buyers' valuations are normalized such that $\sum_j a_{ij} = 1$. It is convenient to write the budgets as a vector $B = (B_i)$ and the valuations as a matrix $A_{n \times m} = (a_{ij})$, such that $A, B$ are the parameters defining the market. We denote the allocation of goods to buyers as a matrix $X = (x_{ij})$ where $x_{ij} \geq 0$ is the (fractional) amount of good

---

[5]Intuitively, if one allows some of the parameters in the dynamic not to update, these parameters become irrelevant, as they will remain frozen, and thus one cannot hope to see any convergence of the entire system.

[6]For the ease of exposition, our proofs use w.l.o.g. $a_{ij} > 0$. This is since in all cases where $a_{ij} = 0$ might have any implication on the proof, such as $\ln(a_{ij})$, these expressions are multiplied by zero in our dynamics.

$j$ that buyer $i$ obtained. We assume w.l.o.g. (by proper normalization) that there is a unit quantity of each good. The price of good $j$ (which depends on the players' actions in the market, as explained below) is denoted by $p_j \geq 0$ and prices are listed as a vector $p = (p_j)$. Buyers have a linear utility function $u_i(x_i) = \sum_j a_{ij} x_{ij}$ with the budget constraint $\sum_j x_{ij} p_j \leq B_i$. We assume w.l.o.g. that the economy is normalized, i.e., $\sum_i B_i = \sum_j p_j = 1$.

**Market equilibrium**: The competitive equilibrium (or "market equilibrium") is defined in terms of allocations and prices as follows.

**Definition 1.** *(Market Equilibrium): A pair of allocations and prices $(X^*, p^*)$ is said to be market equilibrium if the following properties hold:*

1. *Market clearing:* $\quad \forall j, \sum_i x_{ij}^* = 1,$

2. *Budget feasibility:* $\quad \forall i, \sum_j x_{ij}^* p_j^* \leq B_i,$

3. *Utility maximization:* $\quad \forall i, x_i^* \in \arg\max_{x_i} u_i(x_i).$

In words, under equilibrium prices all the goods are allocated, all budgets are used, and no player has an incentive to change their bids given that the prices remain fixed. Notice that this notion of equilibrium is different from a Nash equilibrium of the game where the buyers select their bids strategically, since in the former case, players do not consider the direct effect of possible deviation in their bids on the prices. We discuss this further in Section 3. For linear Fisher markets, it is well established that competitive equilibrium utilities $u^*$ and prices $p^*$ are unique, equilibrium allocations are known to form a convex set, and the following conditions are satisfied.

$$\forall i, j \quad \frac{a_{ij}}{p_j^*} \leq \frac{u_i^*}{B_i} \qquad \text{and} \quad x_{ij} > 0 \implies \frac{a_{ij}}{p_j^*} = \frac{u_i^*}{B_i}.$$

This is a detailed characterization of the equilibrium allocation: every buyer gets a bundle of goods in which all goods maximize the value per unit of money. The quantity $a_{ij}/p_j^*$ is informally known as "bang-per-buck" (ch. 5 & 6 in [58]), the marginal profit from adding a small investment in good $j$.

Market equilibrium bids are also known to maximize the Nash social welfare function (see [31]) $\text{NSW}(X) = \prod_{i \in \mathcal{B}} u_i(x_i)^{B_i}$ and to be Pareto efficient, i.e., no buyer can improve their utility without making anyone else worse off (as stated in the first welfare theorem).

**The trading post mechanism and the market game (Shapley-Shubik)**: First described in [64] and studied under different names [33, 47], the trading post mechanism is an allocation and pricing mechanism which attempts to capture how a price is modified by demand. Buyers place bids on goods, where buyer $i$ places bid $b_{ij}$ on good $j$. Then, the mechanism computes the good's price as the total amount spent on that good and allocates the good proportionally to the bids, i.e., for bids $b$:

$$p_j = \sum_{i=1}^n b_{ij} \qquad x_{ij} = \begin{cases} \frac{b_{ij}}{p_j} & b_{ij} > 0 \\ \\ 0 & \text{otherwise.} \end{cases}$$

Note that the trading post mechanism guarantees market clearing for every bid profile $b$ in which all goods have at least one buyer who is interested in buying. The feasible bid set of a buyer under the budget constraint is $S_i = \{b_i \in \mathbb{R}^m | \forall j \; b_{ij} \geq 0 \quad \sum_j b_{ij} = B_i\}$, i.e., a scaled simplex. Denote $S = \prod_{i \in \mathcal{B}} S_i$ and $S_{-i} = \prod_{k \in \mathcal{B} \setminus \{i\}} S_k$. Considering the buyers as strategic, one can define the *market game* as $G = \{\mathcal{B}, (S_i)_{i \in \mathcal{B}}, (u_i)_{i \in \mathcal{B}}\}$ where the utility functions can be written explicitly as $u_i(b) = u_i(x_i(b)) = \sum_{j=1}^m \frac{a_{ij} b_{ij}}{p_j}$. We sometimes use the notation $u_i(b_i, b_{-i})$, where $b_i$ is the bid vector of player $i$ and $b_{-i}$ denotes the bids of the other players.

**Potential function and Nash equilibrium**: For completeness, we add the following definitions. *Potential function:* A function $\Phi$ is an exact potential function[53] if $\forall i \in \mathcal{B}, \forall b_{-i} \in S_{-i}$ and $\forall b_i, b_i' \in S_i$ we have that $\Phi(b_i', b_{-i}) - \Phi(b_i, b_{-i}) = u_i(b_i', b_{-i}) - u_i(b_i, b_{-i})$, with $u_i$ being $i$'s utility function in the game. *Best response:* $b_i^*$ is a best response to $b_{-i}$ if $\forall b_i \in S_i \quad u_i(b_i^*, b_{-i}) \geq u_i(b_i, b_{-i})$. That is, no other response of $i$ can yield a higher utility. *Nash equilibrium:* $b^*$ is Nash equilibrium if $\forall i \; b_i^*$ is a best response to $b_{-i}^*$ (no player is incentivized to change their strategy).

**Proportional response dynamics**: As explained in the introduction, the proportional response dynamic is specified by an initial bid profile $b^0$, with $b_{ij}^0 > 0$ whenever $a_{ij} > 0$, and the following

update rule for every player that is activated by the adversary: $b_{ij}^{t+1} = \frac{a_{ij}x_{ij}^t}{u_i(x_i^t)}B_i$. See Section 5 for further details on activation of subsets of the players.

## 3   The Associated Game

As mentioned above, the Fisher market can be naturally thought of as a game in which every one of the $n$ players aims to optimize their individual utility $u_i(b_i, b_{-i})$ (see Section 2 for the formal definition). However, it is known that the set of Nash equilibria of this game does not coincide with the set of market equilibria [33, 64], and so a solution to this game (if indeed the players reach a Nash equilibrium) is economically inefficient [12].

A natural question that arises is whether there is some other objective for an individual player that when maximized by all the players, yields the market equilibrium. We answer positively to this question and show that there is a family of utility functions such that in the "associated games" with these utilities for the players, the set of Nash equilibria is identical to the set of market equilibria of the original game (for further details, see also Appendix A).

However, the fact that a Nash equilibrium of an associated game is a market equilibrium still does not guarantee that the players' dynamics will indeed reach this equilibrium. A key element in our proof technique is that we identify, among this family of associated games, a single game, defined by the "associated utility" $\tilde{u}_i(b) = \sum_j b_{ij}\ln(a_{ij}) + \sum_j p_j(1-\ln(p_j))$, which admits an exact potential. We then use a relation which we show between this game and the proportional response update rule to prove the convergence of our dynamics (Theorem 1).

**Definition 2.** *(The Associated Game): Let $G$ be a market game. Define the associated utility of a player $i$ as $\tilde{u}_i(b) = \sum_j b_{ij}\ln(a_{ij}) + \sum_j p_j(1-\ln(p_j))$. The associated game $\tilde{G}$ is the game with the associated utilities for the players and the same parameters as in $G$.*

**Theorem 3.** *For every Fisher market, the associated game $\tilde{G}$ admits an exact potential function that is given by[7] $\Phi(b) = \sum_{ij} b_{ij}\ln(a_{ij}) + \sum_j p_j(1-\ln(p_j))$.*

$\tilde{G}$ is constructed such that the function $\Phi$ is its potential. Note that although having similar structure, $\tilde{u}_i$ and $\Phi$ differ via summation on $i$ only in the first term ($\Phi$ is not the sum of the players' utilities).

Once having the potential function defined, the proof is straightforward: the derivatives of the utilities $\tilde{u}_i$ and the potential $\Phi$ with respect to $b_i$ are equal for all $i$. Theorem 2, formally restated below, connects between the associated game, the market equilibria and the potential.

**Theorem.** *(Restatement of Theorem 2). The following three sets of bid profiles are equal. (1) The set of pure-strategy Nash equilibria of the associated game: $NE(\tilde{G}) = \{b^* \mid \forall b \in S \quad \tilde{u}_i(b^*) \geq \tilde{u}_i(b)\}$; (2) the set of market equilibrium bid profiles of the Fisher market: $\{b^* \mid (x(b^*), p(b^*)) \text{ satisfy Def. 1}\}$; and (3) the maximizing set of the potential from Theorem 3: $\arg\max_{b \in S} \Phi(b)$.*

The proof uses a different associated game $G'$ that has simpler structure than $\tilde{G}$, but does not have an exact potential, and shows that: (i) Nash equilibria of $G'$ are the market equilibria; (ii) all the best responses of players $i$ to bid profiles $b_{-i}$ in $G'$ are the same as those in $\tilde{G}$; and (iii) every equilibrium of $\tilde{G}$ maximizes the potential $\Phi$ (immediate by the definition of potential).

## 4   Best Response Dynamics

In this section we explore another property of the associated game: we show that if instead of using the proportional response update rule, each player myopically plays their best response to the last bid profile with respect to their associated utility, then the entire asynchronous sequence of bids converges to a market equilibrium, as stated in the following theorem. We then show that there is a close relation between best response and proportional response dynamics.

---

[7]Since we discuss the players' associated utilities, we consider maximization of this potential. Of course, if the reader feels more comfortable with minimizing the potential, one can think of the negative function.

**Theorem 4.** *For generic linear Fisher markets in a sequential asynchrony model where in every step a single player is activated, best response dynamics converge to the Market Equilibrium. For non-generic markets the prices are guaranteed to converge to the equilibrium prices.*

The idea of the proof is to show that the best-response functions are single valued ($\forall i, b_{-i} : \tilde{u}_i(\cdot, b_{-i})$ has a unique maximizer) and continuous (using the structure of best-response bids). Together with the existence of the potential function $\Phi$ it holds that the analysis of [46] applies for these dynamics with the associated utilities[8] and thus convergence is guaranteed.

One of the appealing points about proportional response dynamics is their simplicity — in each update, a player observes the obtained utilities and can easily compute the next set of bids. We show that also the best response of a player can be computed efficiently by reducing the calculation to a search over a small part of the subsets of all goods which can be solved by a simple iterative process.

**Proposition 1.** *For every player $i$ and any fixed bid profile $b_{-i}$ for the other players, the best response of $i$ is unique and can be computed in polynomial time.*

Roughly, best responses are characterized uniquely by a one-dimensional variable $c^*$. For every subset of goods $s$ we define a variable $c_s$ and prove that $c^*$ is the maximum amongst all $c_s$. So finding $c^*$ is equivalent to searching a specific subset with maximal $c_s$. The optimal subset of goods admits a certain property that allows to narrow down the search domain from all subsets to only $m$ subsets.

The relation between the best response and proportional response updates can intuitively be thought of as follows. While in PRD players split their budget between all the goods according the utility that each good yields, and so gradually shift more budget to the more profitable subset of goods, best response bids of player $i$ with respect to $\tilde{u}_i$ can be understood as spending the entire budget on a subset of goods which, after bidding so (considering the effect of bids on prices), will jointly have the maximum bang-per-buck (in our notation $a_{ij}/p_j$) amongst *all subsets of goods*, given the bids $b^t_{-i}$ of the other players. Those bids can be regarded as "water-filling" bids as they level the bang-per-buck amongst all goods purchased by player $i$ (for a further discussion see the appendix).

It turns out that there is a clear formal connection between the best response of a player in the associated game and the proportional response update rule in the true game: the best response bids are the limit point of an infinite sequence of proportional response updates by the same player when the bids of the others are held fixed, as expressed in the following proposition.

**Proposition 2.** *Fix any player $i$ and fix any bid profile $b_{-i}$ for the other players. Let $b^*_i = argmax_{b_i \in S_i} \tilde{u}_i(b_i, b_{-i})$ and let $(b^t_i)^\infty_{t=1}$ be a sequence of consecutive proportional response steps applied by player $i$, where $b_{-i}$ is held fixed at all times $t$. Then $\lim_{t\to\infty} b^t_i = b^*_i$.*

## 5 Simultaneous Play by Subsets of Agents

In this section, we shift our focus back to proportional response dynamics under the activation asynchrony model in which the adversary can choose in every step any subset of players to update their bids. Towards proving that proportional response dynamics converges to a market equilibrium in this setting, we utilize the associated game and potential function presented in Section 3 to show that *any activated subset* of players performing a PRD step will increase the potential. Formally, let $v \subseteq \mathcal{B}$ be a subset of players activated by the adversary and let $f_v(b)$ be a function that applies proportional response to members of $v$ and acts as the identity function for all the other players. The update for time $t+1$ when the adversary activates a subset of the players $v^t \subseteq \mathcal{B}$ is therefore:

$$b^{t+1}_{ij} = (f_{v^t}(b^t))_{ij} = \begin{cases} \frac{a_{ij} x^t_{ij}}{u^t_i} B_i & \text{if } i \in v^t \\ b^t_{ij} & \text{otherwise.} \end{cases}$$

**Lemma 1.** *For all $v \subseteq \mathcal{B}$ and for all $b \in S$ it holds that $\Phi(f_v(b)) > \Phi(b)$, unless $f_v(b) = b$.*

The proof shows that for any subset of players $v^t$, a PRD step $b^{t+1}$ is the solution to some maximization problem of a function $g^t(b)$ different from $\Phi$, such that $\Phi(b^{t+1}) > g^t(b^{t+1}) \geq g^t(b^t) = \Phi(b^t)$.

---

[8]Note that while the best-reply function with respect to the standard utility function is formally undefined at zero, the associated utility and its best-reply function are well defined and continuous at zero.

Notable to mention is the sequential case where all subsets are singletons, i.e., for all $t$, $v^t = \{i^t\}$ for some $i^t \in \mathcal{B}$. In that case, the above result yields that the best-response bids can be expressed as the solution to an optimization problem over the bids $b$ on a function that is monotone in the KL divergence between the *prices* induced by $b$ and the current *prices*, whereas PRD is the solution to an optimization problem on a similar function, but one that depends on the KL divergence between the *bids* $b$ and the current *bids*. Thus, sequential PRD can be regarded as a relaxation of best response; on the one hand, it is somewhat simpler to compute a step, and on the other hand, it takes more steps to reach the best response (see Proposition 2 and the simulations in Section 8).

## 6 Convergence of Asynchronous Proportional Response Dynamics

With the results from the previous sections under our belts (namely, the associated game, Theorems 2, 3 about its potential and equilibria, and Lemma 1 about updates by several players simultaneously), we are now ready to complete the proof of Theorem 1 on the convergence of asynchronous proportional response dynamics. We explain here the idea of the proof. The full proof is in the appendix.

**Proof idea of Theorem 1:** Our starting point is that we now know that Proportional Response Dynamics (PRD) steps by subsets of players increase the potential. Therefore, the bids should somehow converge to reach the maximum potential, which is obtained only at the set of market equilibria. Technically, since the sequence of bids $b^t$ is bounded, it must have condensation points. The proof then proceeds by way of contradiction. If the sequence does not converge to the set of equilibrium bid profiles, $ME = \{b^* \mid b^* \text{ is a market equilibrium bid profile}\}$, then there is some subsequence that converges to a bid profile $b^{**}$ outside of this set, which by Theorem 2, must achieve a lower potential than any $b^* \in ME$ (since it is not a market equilibrium, and recall that only market equilibria maximize the potential function).

From this point, the main idea is to show that if the dynamic preserves a "livness" property where the maximum time interval between consecutive updates of a player is bounded by some constant $T$, then the dynamic must reach a point where the bids are sufficiently close to $b^{**}$ such that there must be some future update by some subset of the players under which the potential increases to more than $\Phi(b^{**})$, contradicting the existence of condensation points other than market equilibria (note that the sequence of potential values $\Phi(b^t)$ is increasing in $t$). To show this, the proof requires several additional arguments on the continuity of compositions of PRD update functions that arise under adversarial scheduling, and the impact of such compositions on the potential function. The full proof is in the appendix.

## 7 Generic Markets

Here we show that in the generic case, linear Fisher markets have a unique equilibrium bid profile. While it is well known that in linear Fisher markets equilibrium prices and utilities are unique, and the equilibrium bids and allocations form convex sets (see section 2), we show that multiplicity of equilibrium bid profiles can result only from a special degeneracy in the market parameters that has measure zero in the parameter space. In other words, if the market parameters are not carefully tailored to satisfy a particular equality (formally described below), or, equivalently, if the parameters are slightly perturbed, the market will have a unique equilibrium. Similar property was known for linear exchange markets [8] and we present a simple and concise proof for the Fisher model.

**Definition 3.** *A Fisher market is called generic if the non-zero valuations of the buyers $(a_{ij})$ do not admit any multiplicative equality. That is, for any distinct and non empty $K, K' \subseteq \mathcal{B} \times \mathcal{G}$ it holds that $\prod_{(i,j) \in K} a_{ij} \neq \prod_{(i',j') \in K'} a_{i'j'}$.*

**Theorem 5.** *Every generic linear fisher market has a unique market equilibrium bid profile $b^*$.*

Before discussing the proof of Theorem 5, we have the following corollaries.

**Corollary 1.** *For generic linear Fisher markets, proportional response dynamics with adversarial activation asynchrony, where each player is activated at least once every $T$ steps, converge to the unique market equilibrium.*

The main theorem, Theorem 1, suggests that proportional response dynamics with adversarial activation asynchrony converges to the set of market equilibria. Since the previous theorem guarantees

that a generic market has a unique market equilibrium bid profile, it is clear the dynamics converges to that bid profile.

**Corollary 2.** *In generic linear Fisher markets, no-swap regret dynamics in the associated game converge to the market equilibrium.*

This follows from [56], who showed that in games with convex strategy sets and a continuously differentiable potential function $\Phi$ (as in our case), the set of correlated equilibria consists of mixtures of elements in $\arg\max_b \Phi$. Theorem 2 yields that $\arg\max_b \Phi = b^*$, and so there is a unique correlated equilibrium, which is the market equilibrium, and we have that no-swap regret guarantees convergence to it.

To prove Theorem 5, we use a representation of the bids in the market as a bipartite graph of players and goods $\Gamma(b) = \{V, E\}$ with $V = \mathcal{B} \cup \mathcal{G}$ and $E = \{(i, j) \mid b_{ij} > 0\}$. The proof shows that if a market has more than one equilibrium bid profile, then there has to be an equilibrium $b$ with $\Gamma(b)$ containing a cycle, whereas the following lemma forbids this for generic markets.

**Lemma 2.** *If $b^*$ are equilibrium bids in a generic linear Fisher market, then $\Gamma(b^*)$ has no cycles.*

A key observation for proving this lemma is that at a market equilibrium, for a particular buyer $i$, the quantity $a_{ij}/p_j^*$ is constant amongst goods purchased, and so it is possible to trace a cycle and have all the $p_j^*$ cancel out and obtain an equation contradicting the genericity condition.

An observation that arises from Lemma 2 is that when the number of buyers in the market is of the same order of magnitude as the number of goods or larger, then, in equilibrium, most buyers will only buy a small number of goods. Since there are no cycles in $\Gamma(b^*)$ and there are $n + m$ vertices, there are at most $n + m - 1$ edges. Thus, with $n$ buyers, the average degree of a buyer is $1 + \frac{m-1}{n}$.

## 8 Simulations

Next, we look at simulations of the dynamics that we study and compare the convergence of proportional response dynamics to best response dynamics in the associated game, as discussed in Section 4. The metrics we focus on here for every dynamic are the Nash social welfare [55], which, as mentioned in Section 2, is maximized at the market equilibrium, and the Euclidean distance between the bids at time $t$ and the equilibrium bids. Additionally, we look at the progression over time of the value of the potential $\Phi(b^t)$ (for the definition, see Section 3).

Figure 1 presents simulations of an ensemble of markets, each with ten buyers and ten goods. The parameters in each market (defined in the matrices $A$ and $B$) are uniformly sampled, ensuring that the genericity condition (defined in Section 2) holds with probability one. These parameters are also normalized, as explained in Section 2. For each market, the parameters remain fixed throughout the dynamics. The initial condition in all simulation runs is the uniform distribution of bids over items, and the schedule is sequential, such that a single player updates their bids in each time step.

Figure 1a (main plot) show our metrics for PRD averaged over a sample of 300 such simulations. The insets show the plots of a random sample of 50 individual simulations (without averaging) over a longer time period. Figure 1b show similar plots for best response dynamics.

As could be expected based on our analysis in Section 4, best response dynamics converge faster than PRD, as can be seen in the different time scales on the horizontal axes. A closer look at the individual bid dynamics depicted in the insets reveals a qualitative difference between the two types of dynamics: in PRD, the bids in each dynamic smoothly approach the equilibrium profile, whereas best response bid dynamics are more irregular. Additionally, the collection of curves for the individual simulations shows that under uniformly distributed market parameters, both dynamics exhibit variance in convergence times, with a skewed distribution. In most markets, the dynamics converge quickly, but there is a distribution tail of slower-converging dynamics.

## 9 Conclusion

We have shown that proportional response bid dynamics converge to a market equilibrium in a setting where the schedule of bid updates can be chosen adversarially, allowing for sequential or

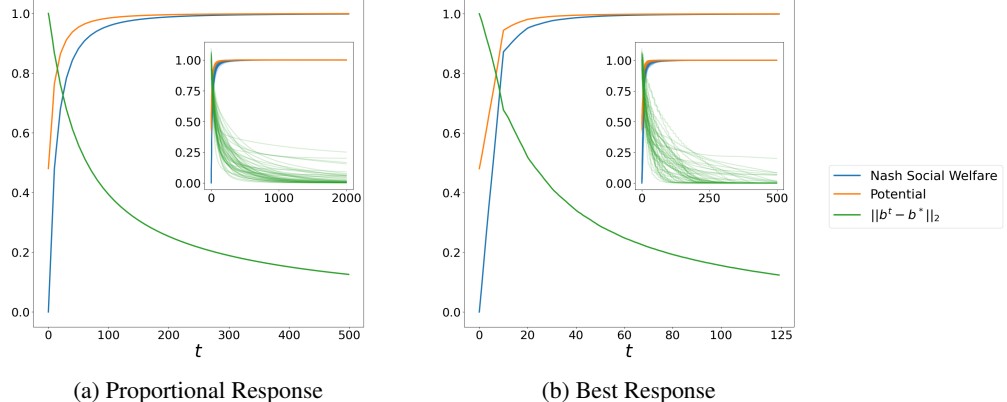

(a) Proportional Response

(b) Best Response

Figure 1: Proportional response and best response dynamics. The main figures show the progression of the average metrics over time and the insets show a collection of individual dynamics over a longer time period.

simultaneous updates for any subset of players. We proposed a novel approach to address this problem by identifying a family of associated games related to proportional response dynamics, showing their relation to the competitive equilibria of the market, and leveraging these relations to prove convergence of the dynamics. En route, we showed that other types of dynamics, such as myopic best response and no swap regret, also converge in the associated game. Additionally, we note that our result on the uniqueness of market equilibria in the generic case (e.g., if the market parameters have some element of randomness) may also be of interest for future research in the Fisher market setting.

One main open question that we did not analyze is whether proportional response dynamics converge under the full asynchrony model, which includes information delays. The analysis of this model raises several complications, as it creates further coupling between past and current bid profiles. We conjecture that if information delays are bounded, then convergence also occurs in this model. However, it is not clear whether our approach could be extended to argue that proportional response updates by subsets of players with respect to delayed information increase the potential in our associated game, or whether proving convergence in this setting will require new methods. One limitation of our analysis is that we provide a guarantee that under any bid update by any subset of players chosen by an adversary, the potential function of the associated game increases, but our technique does not specify by how much the potential increases in every step, and therefore, we do not provide speed of convergence results. Such analysis seems to require new techniques, and we see this as an interesting open problem for future work.

## Acknowledgments

This work was supported by the European Research Council (ERC) under the European Union's Horizon 2020 Research and Innovation Programme (grant agreement no. 740282) and by a grant from the Israeli Science Foundation (ISF number 505/23). During part of this project, Yoav Kolumbus has been affiliated with the Hebrew University of Jerusalem.

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

# Appendices

In the following sections we provide the proofs for the results presented in the main text as well as further technical details and discussion.

In the following, we use $\nabla_{b_i} f$ to denote the gradient of a function $f$ with respect to the bids $b_i$ of player $i$ only. We use $\partial_{b_{ij}} f$ to denote the partial derivative with respect to the bid of player $i$ on good $j$, and $\partial^2_{b_{ij}} f$ to denote the second derivative. We denote by $\theta_i = \sum_{k \neq i} b_i$ as the 'pre-prices,' which represent the prices excluding the bids $b_i$ (and so, for every player $i$ and every bid profile, $p = \theta_i + b_i$). In some of the proofs, we use the abbreviated notation $(f)^+ = \max(f, 0)$. All other notations are as defined in the main text.

## Appendix A   The Associated Game

*Proof.* (Theorem 3): A sufficient condition for $\Phi$ being an exact potential[53] is

$$\forall i \quad \nabla_{b_i} \Phi(b_i, b_{-i}) = \nabla_{b_i} \tilde{u}_i(b_i, b_i).$$

And indeed, in our case we have:

$$\partial_{b_{ij}} \Phi(b_i, b_{-i}) = \ln(a_{ij}) - \ln(p_j) = \ln(\frac{a_{ij}}{p_j}),$$

$$\partial_{b_{ij}} \tilde{u}_i(b_i, b_i) = \ln(a_{ij}) - \ln(p_j) = \ln(\frac{a_{ij}}{p_j}).$$

$\square$

In order to prove Theorem 2, we first define a different associated game denoted $G'$ that differs from $\tilde{G}$ only in having a different associated utility function $u'_i = \sum_j a_{ij} \ln(p_j)$.

In fact, $\tilde{G}$ and $G'$ a part of a family of associated games of the market game $G$, which have the property that they all share the same best responses to bid profiles (and therefore, also the same Nash equilibria) and for all these games, the function $\Phi$ is a best-response potential (see [68] for the definition of best-response potential games). Among this family of games, we are particularly interested in the games $\tilde{G}$ and $G$ since the former admits $\Phi$ as an exact potential, and the latter has a particularly simple derivative for its utility $u'_i$, which has a clear economic interpretation: $\partial_{b_{ij}} u'_i(b) = a_{ij}/p_j$. This is simply the bang-per-buck of player $i$ from good $j$ (see the model section in the main text).

Next, we present several technical lemmas that will assist us in proving Theorem 2 and which will also be useful in our proofs later on.

**Lemma 3.** *For any player $i$ and fixed $b_{-i}$, both $\tilde{u}_i(b_i, b_{-i})$ and $u'_i(b_i, b_{-i})$ are strictly concave in $b_i$.*

*Proof.* We will show the proof for $\tilde{u}_i$. We compute the Hessian and show that it is negative definite. The diagonal elements are

$$\partial^2_{b_{ij}} \tilde{u}_i(b_i, b_i) = -\frac{1}{p_j},$$

and all of the off-diagonal elements are

$$\partial_{b_{ik}} \partial_{b_{ij}} \tilde{u}_i(b_i, b_i) = 0.$$

Therefore, the Hessian is a diagonal matrix with all of its elements being negative, and thus, $\tilde{u}_i$ is strictly concave. The same argument works for $u'_i$ as well. $\square$

**Lemma 4.** *Fix a player $i$ and any bid profile $b_{-i} \in S_{-i}$ of the other players, then the following two facts hold.*

1. $b'^*_i = \arg\max_{b_i \in S_i} u'_i(b_i, b_{-i})$ *if and only if it holds that* $\forall b'_i \in S_i \sum_j \frac{a_{ij}}{p'^*_j} b'^*_{ij} \geq \sum_j \frac{a_{ij}}{p'^*_j} b'_{ij}$, *where* $p'^*_j = \theta_{ij} + b'^*_{ij}$.

2. $\tilde{b}_i^* = \arg\max_{b_i \in S_i} \tilde{u}_i(b_i, b_{-i})$ if and only if it holds that $\forall \tilde{b}_i \in S_i$ $\sum_j \ln(\frac{a_{ij}}{\tilde{p}_j^*})\tilde{b}_{ij}^* \geq \sum_j \ln(\frac{a_{ij}}{\tilde{p}_j^*})\tilde{b}_{ij}$, where $\tilde{p}_j^* = \theta_{ij} + \tilde{b}_{ij}^*$.

*Proof.* We will show the proof for (1), and the proof for (2) is similar. Let $b_i^*$ be a best response to $b_{-i}$ and let $b_i'$ be some other strategy. Consider the restriction of $u_i'(b_i)$ to the line segment $[b_i', b_i^*]$ as follows; define $f(\xi) = u_i'(b_i(\xi))$ for $b_i(\xi) = b_i' + \xi(b_i^* - b_i')$ where $\xi \in [0, 1]$. As $u_i'$ is strictly concave and $b_i^*$ is the unique maximizer of $u_i'$, it holds that $f$ is strictly concave and monotone increasing in $\xi$. Therefore the derivative of $f$ must satisfy at the maximum point $\xi = 1$ that $\frac{d}{d\xi}f(1) \geq 0$. This is explicitly given by

$$\frac{d}{d\xi}f(\xi) = \nabla_{b_i} u_i'(b_i(\xi))(b_i^* - b_i').$$

Therefore, when substituting $\xi = 1$ in the derivative we get $b_i(1) = b_i^*$, and

$$0 \leq \frac{d}{d\xi}f(1) = \nabla_{b_i} u_i'(b_i^*)(b_i^* - b_i')$$
$$= \sum_j \frac{a_{ij}}{p_j^*}b_{ij}^* - \sum_j \frac{a_{ij}}{p_j^*}b_{ij}',$$

which implies $\sum_j \frac{a_{ij}}{p_j^*}b_{ij}' \leq \frac{a_{ij}}{p_j^*}b_{ij}^*$, as required.

To complete the second direction of the proof of (1), consider $b_i^*$ for which the expression stated in right hand side of (1) is true for all $b_i'$. Then, fix any $b_i'$ and again consider the restriction of $u_i'$ to $[b_i', b_i^*]$. By direct calculation, as before but in the inverse direction, it holds that $f(1) \geq 0$, and as $u_i'$ and $f(\xi)$ are strictly concave, it thus must be that $\frac{d}{d\xi}f(\xi)$ is monotone decreasing in $\xi$. Thus, for all $\xi$ we have $\frac{d}{d\xi}f(\xi) \geq \frac{d}{d\xi}f(1) \geq 0$. This must mean that $\xi = 1$ is the maximizer of $f(\xi)$ since for all $\xi$, $\frac{d}{d\xi}f(\xi) \geq 0$ implies that $f(\xi)$ is monotone increasing and therefore $u_i'(b_i^*) \geq u_i'(b_i')$. Finally, note that this holds for any $b_i'$ and hence $b_i^*$ must be a global maximum of $u_i'$.

$\square$

**Lemma 5.** *Let $(c_j)_{j \in [m]} \in \mathbb{R}^m$, if there exists $x^* \in \Delta^m$ (the m-dimensional simplex) such that $\forall x \in \Delta^m$ it holds that $\sum_j c_j x_j \leq \sum_j c_j x_j^* := \alpha$ then:*

1. *for all $j$ we have that $c_j \leq \alpha$, and*

2. *if $x_j^* > 0$ then $c_j = \alpha$.*

*Proof.* (1) Assume for the sake of contradiction that there exists $k$ with $c_k > \alpha$ then $x = e_k$ (the "one-hot" vector with 1 at the $k$'th coordinate and 0 in all other coordinates) yields $\sum_j c_j x_j = c_k > \alpha = \sum_j c_j x_j^*$, a contradiction.

Now, to prove (2), assume for the sake of contradiction that there exists $k$ with $x_k^* > 0$ and $c_k < \alpha \implies c_k x_k^* < \alpha x_k^*$. From (1) we have that $c_j \leq \alpha \implies c_j x_j^* \leq \alpha x_j^*$, summing the strict inequality with the weak ones over all $j$ yields $\sum_j c_j x_j^* < \sum_j \alpha x_j^* = \alpha$, a contradiction. $\square$

**Lemma 6.** *Fix a player $i$ and any bid profile $b_{-i} \in S_{-i}$ of the other players, then the following properties of best-response bids hold in the modified games $\tilde{G}$ and $G'$.*

1. *The support set of $b_i^*$, defined as $s_i^* = \{j | b_{ij}^* > 0\}$, is equal to the set $\{j | a_{ij} > c^* \theta_{ij}\}$, and for every $j \in s_i^*$ we have that $\frac{a_{ij}}{p_j^*} = c^*$.*

2. *Best-response bids with respect to the utilities $u_i'$ and $\tilde{u}_i$ are equal and unique. That is, in the definition from Lemma 4 we have $b_i'^* = \tilde{b}_i^*$ (denoted simply as $b_i^*$).*

3. *Best-response bids are given by $b_{ij}^* = (\frac{a_{ij}}{c^*} - \theta_{ij})^+$ for a unique constant $c^* \in (0, {}^m/_{B_i})$.*

*Proof.* By Lemma 3, $\tilde{u}_i$ and $u_i'$ are strictly concave in $b_i$ for any fixed $b_{-i}$, and so each admits a unique maximizer. To see that they are equal, we use Lemma 4 and introduce constants $c, d$ to obtain

$$\forall b_i \quad \sum_j \frac{a_{ij}}{p_j'^*}\frac{b_{ij}}{B_i} \leq \sum_j \frac{a_{ij}}{p_j'^*}\frac{b_{ij}'^*}{B_i} = c, \text{ and}$$

$$\forall b_i \quad \sum_j \ln(\frac{a_{ij}}{\tilde{p}_j^*})\frac{b_{ij}}{B_i} \leq \sum_j \ln(\frac{a_{ij}}{\tilde{p}_j^*})\frac{\tilde{b}_{ij}^*}{B_i} = d,$$

where $p_j'^* = \theta_{ij} + b_{ij}'^*$ and $\tilde{p}_j^* = \theta_{ij} + \tilde{b}_{ij}^*$.

For the ease of exposition, we assume that $\theta_{ij} > 0$ for all $j$. All the results stated below remain valid also when $\theta_{ij} = 0$ for some $j$.

Proof of (1): Applying Lemma 5 to each of those inequalities (once with $x^* = \frac{1}{B_i}b_i'^*$ and twice with $x^* = \frac{1}{B_i}\tilde{b}_i^*$) and denoting the support sets of $b_{ij}'^*, \tilde{b}_{ij}^*$ as $s'^*, \tilde{s}^*$, respectively, we obtain the following. (1) $\forall j \in s'^*$ we have $\frac{a_{ij}}{p_j'^*} = c$ and $\forall j \notin s'^*$ we have $\frac{a_{ij}}{p_j'^*} \leq c$. Therefore, $\forall j \in s'^*$ bids are positive and $c = \frac{a_{ij}}{p_j'^*} = \frac{a_{ij}}{\theta_{ij}+b_{ij}'^*} < \frac{a_{ij}}{\theta_{ij}}$ while $\forall j \notin s'^*$ the bids are zero, and $\frac{a_{ij}}{\theta_{ij}} \leq \frac{a_{ij}}{\theta_{ij}+0} = \frac{a_{ij}}{p_j'^*} \leq c$, hence $s'^* = \{j | c < \frac{a_{ij}}{\theta_{ij}}\}$. To prove the second inequality, we use the same argument but with $d = \ln(\frac{a_{ij}}{\tilde{p}_j^*})$, and so we have that $\tilde{s}^* = \{j | e^d < \frac{a_{ij}}{\theta_{ij}}\}$.

Proof of (2): We will show that $c = e^d$ and thus obtain that the vectors $b_i'^*, \tilde{b}_i^*$ are identical. Assume by way of contradiction that $c < e^d$, then $j \in \tilde{s}^* \implies c < e^d < \frac{a_{ij}}{\theta_{ij}} \implies j \in s'^*$, i.e., $\tilde{s}^* \subseteq s'^*$. For all $j \in \tilde{s}^*$ it holds that $\frac{a_{ij}}{p_j'^*} = c < e^d = \frac{a_{ij}}{\tilde{p}_j^*} \implies \tilde{p}_j^* < p_j'^* \implies \tilde{b}_{ij}^* < b_{ij}'^*$. Now we sum those inequalities over $\tilde{s}^*$ and extend to the support $s'^*$. By using the subset relation we proved, we obtain a contradiction:

$$B_i = \sum_{j \in \tilde{s}^*} \tilde{b}_{ij}^* < \sum_{j \in \tilde{s}^*} b_{ij}'^* \leq \sum_{j \in s'^*} b_{ij}'^* = B_i.$$

The case where $e^d < c$ follows similar arguments with inverse roles of $s'^*, \tilde{s}^*$. Thus, $c = e^d$ and $s'^* = \tilde{s}^*$, which implies $\frac{a_{ij}}{p_j'^*} = \frac{a_{ij}}{\tilde{p}_j^*}$, meaning that the prices are equal as well for all goods purchased. Therefore $b_i'^* = \tilde{b}_i^*$.

Proof of (3): Finally, observe that for $j \in s'^* \quad c = \frac{a_{ij}}{p_j'^*} = \frac{a_{ij}}{\theta_{ij}+b_{ij}'^*} \implies b_{ij}^* = \frac{a_{ij}}{c^*} - \theta_{ij}$, while otherwise $b_{ij}^* = 0$, and $\frac{a_{ij}}{c^*} - \theta_{ij} \leq 0$. For the bounds on $c$, notice that by definition it is equal to $\frac{u_i^*}{B_i}$ and that $u_i^* \in (0, m)$, as $i$ can receive as little as almost zero (by the definition of the allocation mechanism, if $i$ places a bid on a good it will receive a fraction of this good, no matter how tiny) and receive at most (almost) all the goods.

$\square$

The intuition of the above Lemma 6 is that it shows a property of the structure of best-response bids. If we consider all the goods sorted by the parameter $\frac{a_{ij}}{\theta_{ij}}$, then the best-response bids are characterized by some value $c^*$ which partitions the goods into two parts: goods that can offer the player a bang-per-buck of value $c^*$ and those that cannot. The former set of goods is exactly the support $s^*$. When a player increases its bid on some good $j$, the bang-per-buck offered by that good decreases, so clearly, any good with $c^* \leq \frac{a_{ij}}{\theta_{ij}}$ cannot be considered in any optimal bundle. Consider the situation where the player has started spending money on goods with $\frac{a_{ij}}{\theta_{ij}} > c^*$, and that for some goods $j$ and $k$ we have that $\frac{a_{ij}}{p_j} = \frac{a_{ik}}{\theta_{ik}}$, then if the player increases the bid on $j$ without increasing the bid on $k$, this means that the bids are not optimal since the player could have received higher bang-per-buck by bidding on $k$. The optimal option is a 'water-filling' one: to split the remaining budget and use it to place bids on both $j$ and $k$, yielding equal bang-per-buck for both (as Lemma 6 shows).

With the above lemmas, we are now ready to prove Theorem 2.

*Proof.* (Theorem 2): We start by making the following claim.

*Claim*: The set of Market equilibria is equal to the set of Nash equilibria in the game $G'$.

*Proof.* By definition, $b^*$ is a Nash equilibrium of $G'$ if and only if for every $i$ it holds that $b_i^* = \arg\max_{b_i \in S_i} u_i'(b_i, b_{-i}^*)$, where by Lemma 3, for any fixed $b_{-i}$, the bid profile $b_i^*$ is unique. By Lemma 4, we have that for $x_{ij}^* = b_{ij}^* p_j^*$ and any other $x_{ij}' = b_{ij}' p_j^*$, $\quad u_i'(x_i^*) \geq u_i'(x_i')$, if and only if $(X^*, p^*)$ is a market equilibrium (market clearing and budget feasibility hold trivially). That is, the set of Nash equilibria of the game $G'$ corresponds to the set of market equilibria (i.e., every bid profile $b^*$ which is a market equilibrium must be a Nash equilibrium of $G'$, and vice versa). $\quad\square$

Then, by Lemma 6, best responses by every player $i$ to any bid profile $b_{-i}$ of the other players with respect to $\tilde{u}_i$ and with respect to $u_i'$ are the same. Therefore, every Nash equilibrium in one game must be a Nash equilibrium in the other. Thus, we have that Nash equilibria of the game $\tilde{G}$ are market equilibria, and vice versa – every market equilibrium must be Nash equilibrium of $\tilde{G}$. Finally, at a Nash equilibrium, no player can unilaterally improve their utility, so no improvement is possible to the potential, and in the converse, if the potential is not maximized, then there exists some player with an action that improves the potential, and so by definition their utility function as well, thus contradicting the definition of a Nash equilibrium. Therefore, we have that every bid profile that maximizes the potential is a Nash equilibrium of $\tilde{G}$ and a market equilibrium (and vice versa). $\quad\square$

## Appendix B    Best Response Dynamics

We start with the following characterizations of best-response bids in the games $\tilde{G}$ and $G'$.

**Lemma 7.** *Fix $\theta_i$ and let $b_i^*$ be $i$'s best response to $\theta_i$ with support $s^*$. Define $c_s = \frac{\sum_{j \in s} a_{ij}}{B_i + \sum_{j \in s} \theta_{ij}}$ for every subset $s \subseteq [m]$. Let $c^*$ be as described in Lemma 6. Then, it holds that $c^* = c_{s^*} \geq c_s$ for all $s \subseteq [m]$. Furthermore, if $s^* \not\subseteq s$ then $c_{s^*} > c_s$.*

*Proof.* Let $b_i^*$ be a best response to $\theta_i$ with support $s^*$. By Lemma 6 we have that $b_{ij}^* = (\frac{a_{ij}}{c^*} - \theta_{ij})^+$. By summing over $s^*$ we obtain that $B_i = \sum_{j \in s^*} \frac{a_{ij}}{c^*} - \theta_{ij}$. Rearranging yields $c^* = \frac{\sum_{j \in s^*} a_{ij}}{B_i + \sum_{j \in s^*} \theta_{ij}}$ which is $c_{s^*}$ by definition. Now we prove that $c^* = \max_{s \subseteq [m]} c_s$. A key observation to the proof is that, by Lemma 6, if $j \in s^*$ then $c^* \theta_{ij} < a_{ij}$ and otherwise $c^* \theta_{ij} \geq a_{ij}$.

For a set $s'$ distinct from $s^*$ we have two cases:

*Case (1)*: $s^* \not\subseteq s'$. Consider a bid profile $b_i'$ that for every good $j$ in $s' \cap s^*$ (if the intersection is not empty) places a bid higher by $\epsilon > 0$ than $b_{ij}^*$ and distributes the rest of $i$'s budget uniformly between all other goods in $s$:

$$b_{ij}' = \begin{cases} b_{ij}^* + \epsilon & \text{if } j \in s' \cap s^*, \\ \frac{B_i - \sum_{j \in s' \cap s^*}(b_{ij}^* + \epsilon)}{|s' \setminus s^*|} & \text{otherwise.} \end{cases}$$

For $\epsilon$ small enough, we have $\sum_{j \in s'} b_{ij}' = B_i$ and the support of $b_i'$ is indeed $s'$. For every $j \in s^* \cap s'$ we have $b_{ij}' > b_{ij}^*$ and by adding $\theta_{ij}$ to both sides we obtain $p_j' > p_j^*$; multiplying both sides by $c^*$ yields (i) $c^* p_j' > c^* p_j^* = a_{ij}$, where the equality is by Lemma 6, while for every $j \in s' \setminus s^*$ it holds that $c^* \theta_{ij} \geq a_{ij}$ by which adding $c^* b_{ij}'$ to the left hand side only increases it and implies (ii) $c^* p_j' > a_{ij}$. Summing over inequalities (i) and (ii) for all $j$ appropriately, we obtain $c^* \sum_{j \in s'} p_j' > \sum_{j \in s'} a_{ij}$, observe that $\sum_{j \in s'} p_j' = \sum_{j \in s'}(b_{ij}' + \theta_{ij}) = B_i + \sum_{j \in s'} \theta_{ij}$, and thus by division, we obtain the result: $c^* > \frac{\sum_{j \in s'} a_{ij}}{B_i + \sum_{j \in s'} \theta_{ij}} = c_{s'}$.

*Case (2)*: $s^* \subset s'$. In this case, the idea used above can not be applied since adding $\epsilon$ to every bid $b_{ij}^*$ would create bids $b_{ij}'$ that exceed the budget $B_i$. As stated above, the equality $c^* = \frac{\sum_{j \in s^*} a_{ij}}{B_i + \sum_{j \in s^*} \theta_{ij}}$ holds where the sums are taken over all members of $s^*$, by rearranging we get $c^* B_i + c^* \sum_{j \in s^*} \theta_{ij} = \sum_{j \in s^*} a_{ij}$. For all $j \in s' \setminus s^*$ it holds that $c^* \theta_{ij} \geq a_{ij}$ and by summing those inequalities for all $j$

and adding the equality above we obtain: $c^* B_i + c^* \sum_{j \in s'} \theta_{ij} \geq \sum_{j \in s'} a_{ij}$ Rearranging yields the result: $c^* \geq \frac{\sum_{j \in s'} a_{ij}}{B_i + \sum_{j \in s'} \theta_{ij}} = c_{s'}$.

And so $c^*$ is obtained as the maximum over all $c_s$, as required. $\square$

**Lemma 8.** *The function $BR_i : S_{-i} \to S_i$ which maps $b_{-i}$ to its best response $b_i^*$ is continuous.*

*Proof.* By Lemma 6, best-response bids are given by $b_{ij}^* = \max\{\frac{a_{ij}}{c^*} - \theta_{ij}, 0\}$, with support $s_i^*$. We wish to show that $b_i^*$ is a continuous in $b_{-i}$. We do so by showing that $b_{ij}^*$ is obtained by a composition of continuous functions. As $\theta_i$ is a sum of elements from $b_{-i}$, it suffices to prove continuity in the variable $\theta_i$. The expression for $b_{ij}^*$ is the maximum between zero and a continuous function of $\theta_{ij}$, which is continuous in $\theta_i$, and so we are left to prove that $\frac{a_{ij}}{c^*} - \theta_{ij}$ is continuous in $\theta_i$. More specifically, it suffices to show that $c^*$ as defined in Lemma 6 is continuous in $\theta_i$.

By Lemma 7, $c^*$ is obtained as the maximum over all $c_s$ functions, where each is a continuous function itself in $\theta_i$, and thus $c^*$ is continuous in $\theta_i$. $\square$

To prove Theorem 1 on the convergence of best-response dynamics we use the following known result (for further details see [46]).

*Theorem (Jensen 2009 [46]):* Let $G$ be a best-reply potential game with single-valued, continuous best-reply functions and compact strategy sets. Then any admissible sequential best-reply path converges to the set of pure strategy Nash equilibria.

*Proof.* (Theorem 4): $\tilde{G}$ is a potential game, which is a stricter notion than being a best-reply potential game (i.e., every potential game is also a best-reply potential game). By Lemma 6, best replies are unique, and so the function $BR_i$ is single valued. Furthermore, Lemma 8 shows that it is also a continuous function. By definition, for every $i$ the strategy set $S_i$ is compact, and so their product $S$ is compact as well. Note that while the best-reply function with respect to the standard utility function is formally undefined when the bids of all other players on some good $j$ are zero, what we need is for the best-reply function to the associated utility to be continuous, and this is indeed the case. Admissibility of the dynamics is also guaranteed by the liveness constraint on adversarial scheduling of the dynamics, and thus by the theorem cited above, best-reply dynamics converges to the set of Nash equilibria of $\tilde{G}$. Since every element in this set is market equilibrium (by Theorem 2) and equilibrium prices are unique (see the model section in the main text), we have that any dynamic of the prices are guaranteed to converge to equilibrium prices. Furthermore for a generic market there is a unique market equilibrium (by Theorem 5) and convergence to the set in fact means convergence to the point $b^*$, the market-equilibrium bid profile. $\square$

*Proof.* (Proposition 1): Fix a player $i$, fix any bid profile $b_{-i}$ of the other players and let $b_i^*$ be $i$'s best response to $b_{-i}$, by Lemma 6, $b_{ij}^* = (\frac{a_{ij}}{c^*} - \theta_{ij})^+$ for $c^*$ being a unique constant. We present a simple algorithm which computes $c^*$ and has a run-time of $\mathcal{O}(m \log(m))$.

---

**Algorithm 1** Compute $c^*$

**Require:** $a_i, B_i, \theta_i$

Sort the values $a_i, \theta_i$ according to $\frac{a_{ij}}{\theta_{ij}}$ in a descending order. If there are goods with $\theta_{ij} = 0$, sort them separately according to $a_{ij}$ and place them as a prefix (lower indices) before the other sorted goods. Equal values are sorted in a lexicographical order.

Set: $a \leftarrow 0, \quad \theta \leftarrow 0, \quad c_s \leftarrow 0, \quad c^* \leftarrow 0$
**for** $j = 1, \ldots, m$ **do**
    $a \leftarrow a + a_{ij}, \; \theta \leftarrow \theta + \theta_{ij}$
    $c_s \leftarrow \frac{a}{\theta + B_i}$
    $c^* \leftarrow \max\{c^*, c_s\}$
**end for**
**return** $c^*$

---

To see that this process indeed reaches $c^*$, assume w.l.o.g. that the goods are sorted by $\frac{a_{ij}}{\theta_{ij}}$ in a descending order. For ease of exposition, assume $\theta_{ij} > 0$ for all $j$; the case with $\theta_{ij} = 0$ for some goods is similar. By Lemma 6 we have $s^* = \{j | a_{ij} > c^* \theta_{ij}\}$. And so, if $k < j$ and $j \in s^*$ then $k \in s^*$, since in this case $\frac{a_{ik}}{\theta_{ik}} > \frac{a_{ij}}{\theta_{ij}} > c^*$. Therefore, $s^*$ must be one of the following sets: $[1], [2], [3], \ldots, [m]$. By Lemma 7 we have $c^* = \max_{s \subseteq [m]} c_s$. For any set mentioned, the algorithm computes $c_s = \frac{\sum_{j \in s} a_{ij}}{B_i + \sum_{j \in s} \theta_{ij}}$ and finds the maximal among all such $c_s$, and therefore it finds $c^*$.

As for the running time of the algorithm, it is dominated by the running time of the sorting operation which is $\mathcal{O}(m \log(m))$. $\qquad \square$

After proving that the best response to a bid profile can be computed efficiently, we can prove now that proportional response, applied by a single player while all the other players' bids are held fix, converges in the limit to that best response.

*Proof.* (Proposition 2): Fix a player $i$ and fix any bid profile $b_{-i}$ of the other players, let $b_i^*$ be the best response of $i$ to $b_{-i}$ with support $s^*$ and let $(b_i^t)_{t=1}^\infty$ be a sequence of consecutive proportional responses made by $i$. That is, $b_i^{t+1} = f_i(b_i^t)$. We start the proof with several claims proving that any sub-sequence of $(b_i^t)_{t=1}^\infty$ cannot converge to any fixed point of $f_i$ other than $b_i^*$. After establishing this, we prove that the sequence indeed converges to $b_i^*$.

*Claim 1*: Every fixed point of Proportional Response Dynamic has equal 'bang-per-buck' for all goods with a positive bid. That is, if $b_i^{**}$ is a fixed point of $f_i$ then $\frac{a_{ij}}{p_j^{**}} = \frac{u_i^{**}}{B_i}$ for every good $j$ with $b_{ij}^{**} > 0$, where $u_i^{**}$ is the utility achieved for $i$ with the bids $b_i^{**}$.

*Proof*: By substituting $b_i^{**}$ into the PRD update rule, we have

$$b_i^{**} = f_i(b_i^{**})$$
$$\iff \forall j \quad b_{ij}^{**} = \frac{a_{ij}/p_j^{**}}{u_i^{**}/B_i} b_{ij}^{**}$$
$$\iff \text{either } b_{ij}^{**} = 0 \text{ or } a_{ij}/p_j^{**} = u_i^{**}/B_i.$$

$\qquad \square$

*Claim 2*: The following properties of $b_i^*$ hold.

1. Except $b_i^*$, there are no other fixed points of $f_i$ with a support that contains the support of $b_i^*$. Formally, there are no fixed points $b_i^{**} \neq b_i^*$ of $f_i$ with support $s^{**}$ such that $s^* \subset s^{**}$.

2. The bids $b_i^*$ achieve a higher utility in the original game $G$, denoted $u_i^*$, than any other fixed point of Proportional Response Dynamics. Formally, let $b_i^{**}$ be any fixed point other than $b_i^*$, with utility $u_i^{**}$ in the original game $G$, then $u_i^* > u_i^{**}$.

*Proof*: Let $b_i$ be any fixed point of $f_i$. By the previous claim it holds that $\frac{a_{ij}}{p_j} = \frac{u_i}{B_i}$ whenever $b_{ij} > 0$. Multiplying by $p_j$ yields $a_{ij} = \frac{u_i}{B_i} p_j$. Summing over $j$ with $b_{ij} > 0$ and rearranging yields $\frac{u_i}{B_i} = \frac{\sum_{j \in s} a_{ij}}{\sum_{j \in s} p_j} = c_s$ as defined in Lemma 7 with support $s$. By that lemma, we have that $c_{s^*} \geq c_s$ for any set $s$ distinct from $s^*$. Thus, we have that $u_i^*/B_i = c_{s^*} \geq c_{s^{**}} = u_i^{**}/B_i$ for $b_i^{**}$ being a fixed point of $f_i$ other than $b_i^*$ with support $s^{**}$ and utility value $u_i^{**}$.

Assume for the sake of contradiction that $s^* \subset s^{**}$. If $j \in s^*$ then $j \in s^{**}$. By Claim 1 for every such $j$ the following inequality holds,

$$\frac{a_{ij}}{p_j^*} = \frac{u_i^*}{B_i} \geq \frac{u_i^{**}}{B_i} = \frac{a_{ij}}{p_j^{**}},$$

implying that $p_j^{**} \geq p_j^*$. Subtracting $\theta_{ij}$ from both sides yields $b_{ij}^{**} \geq b_{ij}^*$. Summing over $j \in s^*$ yields a contradiction:

$$B_i = \sum_{j \in s^*} b_{ij}^* \leq \sum_{j \in s^*} b_{ij}^{**} < \sum_{j \in s^{**}} b_{ij}^{**} = B_i,$$

where the first inequality is as explained above, and the last by the strict set containment $s^* \subset s^{**}$.

Finally, as there are no fixed points with support $s^{**}$ containing $s^*$, by Lemma 7, the inequality stated above is strict, that is $c_{s^*} > c_{s^{**}}$ and so $u_i^* > u_i^{**}$.

$\square$

*Claim 3*: If $b_i^{**} \neq b_i^*$ is a fixed point of $f_i$ then $b_i^{**}$ is not a limit point of any sub-sequence of $(b_i^t)_{t=0}^{\infty}$.

*Proof*: The proof considers two cases: (1) When $u_i$ is continuous at $b^{**}$ (2) when continuity doesn't hold. Let $(b_i^{t_k})_{k=1}^{\infty}$ be a converging subsequence of $(b_i^t)_{t=0}^{\infty}$.

*Case (1)*: The utility function $u_i$ is continuous at $b_i^{**}$ when for every good $j$ it holds that $\theta_{ij} > 0$ or $b_{ij}^{**} > 0$. i.e. that there is no good $j$ with both $\theta_{ij} = 0$ and $b_{ij}^{**} = 0$. This is implied directly from the allocation rule $x_{ij} = \frac{b_{ij}}{\theta_{ij} + b_{ij}}$ (see the formal definition in Section 2) and the fact that $u_i = \sum_j a_{ij} x_{ij}$. Examine the support of $b_i^{**}$, by Claim 2 there are no fixed points with support set $s^{**}$ containing $s^*$. Therefore $s^* \not\subset s^{**}$ implying that there exists a good $j \in s^* \setminus s^{**}$. That is, by definition of the supports, there exists $j$ with $b_{ij}^* > 0$ and $b_{ij}^{**} = 0$. Consider such $j$ and assume for the sake of contradiction that $b_i^{**}$ is indeed a limit point. Then, by definition, for every $\delta^{**} > 0$ exists a $T$ s.t. if $t > T$ then $\|b_i^{t_k} - b_{ij}^{**}\| < \delta^{**}$. Specifically it means that $|b_{ij}^{t_k} - b_{ij}^{**}| < \delta^{**}$ whenever $t > T$.

By Claim 2, $u_i^* > u_i^{**}$. Then, by continuity there exists a $\delta'$ s.t. if $\|b_i^{**} - b_i\| < \delta'$ then $|u_i(b_i) - u_i^{**}| < u_i^* - u_i^{**}$. Take $\delta^{**} < \min\{\delta', b_{ij}^*\}$ and, by the assumption of convergence, there is a $T$ s.t. for $t > T$ and we have that $\|b_i^{t_k} - b_i^{**}\| < \delta^{**}$. This implies (I) $|b_{ij}^{t_k} - 0| < \delta^{**} < b_{ij}^*$ as $b_{ij}^{**} = 0$ and (II) $|u_i^{t_k} - u_i^{**}| < u_i^* - u_i^{**} \implies u_i^{t_k} < u_i^*$. From these two, we can conclude that

$$
\frac{a_{ij}}{p_j^{t_k}} = \frac{a_{ij}}{\theta_{ij} + b_{ij}^{t_k}} > \frac{a_{ij}}{\theta_{ij} + b_{ij}^*} = \frac{u_i^*}{B_i} > \frac{u_i^{t_k}}{B_i}.
$$

Finally, observe that by rearranging the PRD update rule we get $b_{ij}^{t+1} = \frac{a_{ij}/p_j^{t_k}}{u_i^{t_k}/B_i} b_{ij}^{t_k}$, implying that $b_{ij}^{t_k+1} > b_{ij}^{t_k}$ since $\frac{a_{ij}/p_j^{t_k}}{u_i^{t_k}/B_i} > 1$ for $t > T$ and $b_{ij}^0 > 0$. This means that for all $t_k > T$ we have $b_{ij}^{t_k} > b_{ij}^{T+1}$. That is, $b_{ij}^{t_k}$ cannot converge to zero and thus the subsequence cannot converge to $b_i^{**}$, a contradiction.

*Case (2)*: When there exists a good $j$ with $\theta_{ij} = 0$ and $b_{ij}^{**} = 0$ we have that $u_i$ is not continuous at $b_i^{**}$ and the previous idea doesn't work. Instead we will contradict the PRD update rule. Assume of the sake of contradiction that $b_i^{**}$ is a limit point of a subsequence of PRD updates. Therefore for every $\epsilon$ exists a $T$ s.t. if $t_k > T$ then $|b_{ij}^{t_k} - b_{ij}^{**}| < \epsilon$. Note that $b_{ij}^{**} = 0$ in this case and set $\epsilon < \frac{a_{ij}}{m} B_i$ and so, for $t_k > T$ it holds that $\frac{a_{ij}}{b_{ij}^{t_k}} > \frac{m}{B_i}$. Also note that $p_j^{t_k} = \theta_{ij}^{t_k} + b_{ij}^{t_k} = b_{ij}^{t_k}$ and that the maximal utility a buyer may have is $m$ (when it is allocated every good entirely). Then overall we have that $\frac{a_{ij}}{p_j^{t_k}} > \frac{m}{B_i} > \frac{u_i^{t_k}}{B_i}$. The PRD update rule is $b_{ij}^{t_k+1} = \frac{a_{ij}/p_j^{t_k}}{u_i^{t_k}/B_i} b_i^{t_k}$. But since the ratio $\frac{a_{ij}/p_{ij}^{t_k}}{u_i^{t_k}/B_i}$ is greater than 1 it must be that $b_{ij}^{t_k+1} > b_{ij}^{t_k}$. And so every subsequent element of the subsequence is bounded below by $b_{ij}^{T+1} > 0$ and as before, we reach a contradiction as the subsequence cannot converge to $b_i^{**}$.

$\square$

Finally we can prove the convergence of the sequence $(b_i^t)_{t=1}^{\infty}$. As the action space $S_i$ is compact, there exists a converging subsequence $b_i^{t_k}$ with the limit $b_i^{**}$. If $b_i^{**} = b_i^*$ for any such subsequence, then we are done. Otherwise, assume $b_i^{**} \neq b_i^*$. By the previous claim any fixed point of $f_i$ other than $b_i^*$ is not a limit point of any subsequence, thus $b_i^{**}$ is not a fixed point of $f_i$. By Lemma 1, any subset of players performing proportional response, strictly increase the potential function unless performed at a fixed point. When discussing a proportional response of a single player, with all others remaining fixed, this implies, by the definition of potential function, that $\tilde{u}_i$ is increased at each such step. Let $\epsilon < \tilde{u}_i(f_i(b^{**})) - \tilde{u}_i(b^{**})$, this quantity is positive since $b_i^{**}$ is not a fixed point. The function $\tilde{u}_i \circ f_i$ is a continuous function and $b_i^{t_k}$ converges to $b_i^{**}$ therefore there exists a $T$ such that for all $t_k > T$ we have that $|\tilde{u}_i(f_i(b^{**})) - \tilde{u}_i(f_i(b^{t_k}))| < \epsilon$. Substituting $\epsilon$ yields $\tilde{u}_i(f_i(b^{**})) - \tilde{u}_i(f_i(b^{t_k})) < \tilde{u}_i(f_i(b^{**})) - \tilde{u}_i(b^{**})$ which implies $\tilde{u}_i(b^{**}) < \tilde{u}_i(f_i(b^{t_k})) = \tilde{u}_i(b^{t_k+1}) \leq \tilde{u}_i(b^{t_{k+1}})$. That is, the

sequence $u_i(b_i^{t_k})$ is bounded away from $\tilde{u}_i(b^{**})$ and since $\tilde{u}_i$ is a continuous function, this implies that $b_i^{t_k}$ is bounded away from $b_i^{**}$ — a contradiction to convergence. $\qquad\square$

## Appendix C   Simultaneous Play by Subsets of Agents

In order to prove Lemma 1, we first need some further definitions and technical lemmas. We use the notation $D(x\|y)$ to denote the KL divergence between the vectors $x$ and $y$, i.e., $D(x\|y) = \sum_j x_j \ln(\frac{x_j}{y_j})$. For a subset of the players $v \subseteq \mathcal{B}$, the subscript $v$ on vectors denotes the restriction of the vector to the coordinates of the players in $v$, that is, for a vector $b$ we use the notation $b_v = (b_{ij})_{i\in v, j\in[m]}$ to express the restriction to the subset. $\ell_\Phi(b_v; b_v')$ denotes the linear approximation of $\Phi$; that is, $\ell_\Phi(b_v; b_v') = \Phi(b_v') + \nabla_{b_v}\Phi(b_v')(b_v - b_v')$.

The idea described in the next lemma to present the potential function as a linear approximation term and a divergence term was first described in [5] for a different scenario when all agents act together in a synchronized manner using mirror descent; we extend this idea to our asynchronous setting which requires using different methods and as well as embedding it in a game.

**Lemma 9.** *Fix a subset of the players $v \subset \mathcal{B}$ and a bid profile $b_{-v}$ of the other players. Then, for all $b_v, b_v' \in S_v$ we have that $\Phi(b_v) = \ell_\Phi(b_v; b_v') - D(p\|p')$, where $p = \sum_{i\notin v} b_{ij} + \sum_{i\in v} b_{ij}$ and $p' = \sum_{i\notin v} b_{ij} + \sum_{i\in v} b_{ij}'$.*

*Proof.* Calculating the difference $\Phi(b_v) - \ell_\Phi(b_v; b_v')$ yields
$$\Phi(b_v) - \ell_\Phi(b_v; b_v') = \Phi(b_v) - \Phi(b_v') - \nabla_{b_v}\Phi(b_v')(b_v - b_v').$$

We rearrange the term $\Phi(b_v) - \Phi(b_v')$ as follows.
$$\Phi(b_v) - \Phi(b_v') = \sum_{i\in v, j\in[m]} (b_{ij} - b_{ij}')\ln(a_{ij}) - \sum_j (p_j \ln(p_j) - p_j' \ln(p_j')) - \sum_j (p_j - p_j')$$
$$= \sum_{i\in v, j\in[m]} (b_{ij} - b_{ij}')\ln(a_{ij}) - \sum_j (p_j \ln(p_j) - p_j' \ln(p_j')),$$

where the last equality is since $\sum_j p_j = 1$ for any set of prices because the economy is normalized (see the model section in the main text).

The term $\nabla_{b_v}\Phi(b_v')(b_v - b_v')$ is expanded as follows.
$$\nabla_{b_v}\Phi(b_v')(b_v - b_v') = \sum_{i\in v, j\in[m]} \ln(\frac{a_{ij}}{p_j'})(b_{ij} - b_{ij}')$$
$$= \sum_{i\in v, j\in[m]} \ln(a_{ij})(b_{ij} - b_{ij}') - \sum_{i\in v, j\in[m]} \ln(p_j')(b_{ij} - b_{ij}').$$

Subtracting the latter from the former cancels out the term $\sum_{i\in v, j\in[m]} \ln(a_{ij})(b_{ij} - b_{ij}')$, and we are left with the following.

$$\Phi(b_v) - \ell_\Phi(b_v; b_v') = \Phi(b_v) - \Phi(b_v') - \nabla_{b_v}\Phi(b_v')(b_v - b_v')$$
$$= \sum_{i\in v, j\in[m]} \ln(p_j')(b_{ij} - b_{ij}') - \sum_j (p_j \ln(p_j) - p_j' \ln(p_j'))$$
$$= -\sum_j p_j \ln(p_j) - (p_j' - \sum_{i\in v} b_{ij}' + \sum_{i\in v} b_{ij})\ln(p_j')$$
$$= -\sum_j p_j \ln(p_j) - (\theta_{vj} + \sum_{i\in v} b_{ij})\ln(p_j')$$
$$= -\sum_j p_j \ln(p_j) - p_j \ln(p_j')$$
$$= -D(p\|p').$$

$\qquad\square$

**Lemma 10.** *For any subset of the players $v \subset \mathcal{B}$ and any bid profile $b_{-v}$ of the other players and for every $b_v, b'_v \in S_v$ it holds that $D(p\|p') \leq D(b_v\|b'_v)$, with equality only when $b_v = b'_v$.*

*Proof.* We begin by proving a simpler case where $v = \{i\}$ for some player $i$ and use it to prove the more general statement. Fix $i$ and $b_{-i}$, which implies fixing some $\theta_i$. KL divergence is convex in both arguments with equality only if the arguments are equal; formally, for $\lambda \in (0,1)$ it holds that $D(\lambda\theta_i + (1-\lambda)b_i \| \lambda\theta_i + (1-\lambda)b'_i) \leq \lambda D(\theta_i\|\theta_i) + (1-\lambda)D(b_i\|b'_i)$, which is equivalent to $D(\lambda\theta_i + (1-\lambda)b_i \| \lambda\theta_i + (1-\lambda)b'_i) \leq (1-\lambda)D(b_i\|b'_i)$, with equality only if $b_i = b'_i$ (since $D(\theta_i\|\theta_i) = 0$). Substituting $\lambda = \frac{1}{2}$ and noting that $p_j = \theta_{ij} + b_{ij}$ (and the same for $p'_j$ and $b'_{ij}$), we obtain the following relation.

$$D(\frac{1}{2}p\|\frac{1}{2}p') = D(\frac{1}{2}\theta_i + \frac{1}{2}b_i \| \frac{1}{2}\theta_i + \frac{1}{2}b'_i)$$
$$\leq \frac{1}{2}D(b_i\|b'_i).$$

On the other hand, the expression $D(\frac{1}{2}p\|\frac{1}{2}p')$ can be evaluated as follows.

$$D(\frac{1}{2}p\|\frac{1}{2}p') = \sum_j \frac{1}{2}p_j \ln(\frac{^{1}/_{2}p_j}{^{1}/_{2}p'_j})$$
$$= \frac{1}{2}\sum_j p_j \ln(\frac{p_j}{p'_j})$$
$$= \frac{1}{2}D(p\|p').$$

And therefore, we have $D(p\|p') \leq D(b_i\|b'_i)$, with equality only if $b_i = b'_i$.

Now we can prove the general case, as stated fix $v$ and $b_{-v}$ and let $b_v, b'_v \in S_v$. We know that for all $i \in v$ it is true that $D(p\|p') \leq D(b_i\|b'_i)$, summing those inequalities for all $i \in v$ yields $|v|D(p\|p') \leq \sum_{i\in v} D(b_i\|b'_i)$, on the one hand clearly $D(p\|p') \leq |v|D(p\|p')$ and on the other hand $\sum_{i\in v} D(b_i\|b'_i) = \sum_{i\in v}\sum_j b_{ij}\ln(\frac{b_{ij}}{b'_{ij}}) = D(b_v\|b'_v)$ and the result is obtained. $\square$

**Lemma 11.** *Let $v \subseteq \mathcal{B}$, let $f_v : S \to S$ be a proportional response update function for members of $v$ and identity for the others, and let $b' \in S$ be some bid profile. Then, $(f_v(b'))_v = \arg\max_{b_v \in S_v}\{\ell_\Phi(b_v; b'_v) - D(b_v\|b'_v)\}$.*

*Proof.* By adding and removing constants that do not change the maximizer of the expression on the right hand side, we obtain that the maximizer is exactly the proportional response update rule:

$$\arg\max_{b_v \in S_v}\{\ell_\Phi(b_v; b'_v) - D(b_v\|b'_v)\} = \arg\max_{b_v \in S_v}\{\Phi(b'_v) + \nabla_{b_v}\Phi(b'_v)(b_v - b'_v) - D(b_v\|b'_v)\}$$
$$= \arg\max_{b_v \in S_v}\{\nabla_{b_v}\Phi(b'_v)b_v - D(b_v\|b'_v)\}$$
$$= \arg\max_{b_v \in S_v}\{\nabla_{b_v}\Phi(b'_v)b_v - D(b_v\|b'_v) - \sum_{i\in v} B_i\ln(\frac{u'_i}{B_i})\}.$$

Rearranging the last expression by elements yields the following result,

$$\nabla_{b_v}\Phi(b'_v)b_v - D(b_v\|b'_v) - \sum_{i\in v} B_i \ln(\frac{u'_i}{B_i}) =$$

$$= \sum_{i\in v, j\in[m]} b_{ij}\ln(\frac{a_{ij}}{p'_j}) - \sum_{i\in v, j\in[m]} b_{ij}\ln(\frac{b_i j}{b'_{ij}}) - \sum_{i\in v, j\in[m]} b_{ij}\ln(\frac{u'_i}{B_i})$$

$$= \sum_{i\in v, j\in[m]} b_{ij}\ln(\frac{a_{ij}}{p'_j}\frac{b'_{ij}}{b_{ij}}\frac{B_i}{u'_i})$$

$$= \sum_{i\in v, j\in[m]} b_{ij}\ln(\frac{\frac{a_{ij}x'_{ij}}{u'_i}B_i}{b_{ij}})$$

$$= -\sum_{i\in v, j\in[m]} b_{ij}\ln(\frac{b_{ij}}{\frac{a_{ij}x'_{ij}}{u'_i}B_i}),$$

which is exactly $-D(b_v\|(f_v(b'))_v)$, since $(f_v(b'))_{ij} = \frac{a_{ij}x'_{ij}}{u'_i}B_i$ for $i\in v$ by definition. That is, our maximization problem is equivalent to $\arg\min\{D(b_v\|(f_v(b'))_v)\}$. Finally, note that KL divergence is minimized when both of its arguments are identical, and $(f_v(b'_v))_v \in S_v$, the domain of the minimization. □

*Proof.* (Lemma 1): Let $v\subseteq\mathcal{B}$ be a subset of players and let $b\in S$ be some bid profile. By combining the lemmas proved in this section have that

$$\Phi(f_v(b)) \geq \ell_\Phi(f_v(b); b) - D(f_v(b)\|b) \geq \ell_\Phi(b; b) - D(b\|b) = \Phi(b),$$

where the first inequality is by Lemmas 9 and 10 with the inequality being strict whenever $f_v(b)\neq b$, and the second inequality is by Lemma 11, as $f_v(b)$ was shown to be the maximizer of this expression over all $b\in S$. □

An interesting case to note here is when $v = i$. In this case, the lemmas above show that if the players' bids are $b^t$ and $i$ is being activated by the adversary, then the best response bids of $i$ to $b^t_{-i}$ are the solutions to the optimization problem $\arg\max_{b_i\in S_i}\{\ell_{\tilde{u}_i}(b_i; b^t_i) - D(p\|p^t)\}$. On the other hand, the proportional response to $b^t_{-i}$ is the solution to the optimization problem $\arg\max_{b_i\in S_i}\{\ell_{\tilde{u}_i}(b_i; b^t_i) - D(b_i\|b^t_i)\}$. This can be seen as a relaxation of the former, as proportional response does not increase $\tilde{u}_i$ (or equivalently the potential) as much as best response does. However, proportional response is somewhat easier to compute.

## Appendix D Convergence of Asynchronous Proportional Response Dynamics

*Proof.* (Theorem 1): Denote the distance between a point $x$ and a set $S$ as $d(x, S) = \inf_{x^*\in S}\|x - x^*\|$. By Theorem 2 we have that the set of potential maximizing bid profiles is identical to the set of market equilibria. Denote this set by $ME$ and the maximum value of the potential by $\Phi^*$. More specifically, every $b^*\in ME$ achieves $\Phi(b^*) = \Phi^*$, and $\Phi^*$ is achieved only by elements in $ME$.

We start with the following lemma.

**Lemma 12.** *For every $\epsilon > 0$ there exists a $\delta > 0$ such that $\Phi(b) > \Phi^* - \delta$ implies $d(b, ME) < \epsilon$.*

*Proof.* Assume otherwise that for some $\epsilon_0$ there exist a sequence $(b_t)$ such that $\Phi(b_t)\to\Phi^*$ but $\inf_{b^*\in ME}\|b_t - b^*\|\geq\epsilon_0$ for all $t$. Note that for all $b^*\in ME$ we have that $\Phi(b^*) = \Phi^*$ and $\|b_t - b^*\|\geq\inf_{b'\in ME}\|b_t - b'\|\geq\epsilon_0$ for all $t$. Take a condensation point $b^{**}$ of this sequence and a subsequence $(t_j)$ that converges to $b^{**}$. Thus by our assumption we have $\Phi(b^{**}) = \lim\Phi(b_{t_j}) = \Phi^*$. For any $b^*\in ME$ we have $\|b^{**} - b^*\| = \lim\|b_{t_j} - b^*\|\geq\epsilon_0 > 0$. The former equality must imply $b^{**}\in ME$, but the latter implies $b^{**}\notin ME$. □

Next, for a subset of players $A \subset \mathcal{B}$ let $f_A : [0,1]^n \to [0,1]^n$ be the continuous function where $i \in A$ do a proportional response update and the other players play the identity function. (i.e., do not change their bids, see Section 5 in the main text).

By Lemma 1 from the main text we have that (i) For all $A$ we have that $f_A(b) = b$ if and only if for all $i \in A$ it holds that $f_i(b) = b$; and (ii) $\Phi(f_A(b)) > \Phi(b)$ unless $f_A(b) = b$.

**Definition.** *The stable set of $b^{**}$ is defined to be $S(b^{**}) = \{i | f_i(b^{**}) = b^{**}\}$.*

A corollary (i) and (ii) above is that if $A \subseteq S(b^{**})$ then $f_A(b^{**}) = b^{**}$, but if $A \setminus S(b^{**}) \neq \emptyset$ then $\Phi(f_A(b^{**})) > \Phi(b^{**})$.

**Lemma 13.** *: Let $\Phi(b^{**}) < \Phi^*$. Then there exists $\delta > 0$ such that for every $\|b - b^{**}\| \leq \delta$ and every $A \setminus S(b^{**}) \neq \emptyset$ we have that $\Phi(f_A(b)) > \Phi(b^{**})$.*

*Proof.* Fix a set $A$ such that $A \setminus S(b^{**}) \neq \emptyset$ and let $\alpha = \Phi(f_A(b^{**})) - \Phi(b^{**}) > 0$. Since $\Phi(f_A(\cdot))$ is continuous, there exists $\delta$ so that $|b - b^{**}| \leq \delta$ implies $\Phi(f_A(b^{**})) - \Phi(f_A(b)) < \alpha$ and thus $\Phi(f_A(b)) > \Phi(b^{**})$. Now take the minimum $\delta$ for all finitely many $A$. $\square$

**Lemma 14.** *Let $\Phi(b^{**}) < \Phi^*$ and let $F$ be a finite family of continuous functions such that for every $f \in F$ we have that $f(b^{**}) = b^{**}$. Then there exists $\epsilon > 0$ such that for every $b$ such that $\|b - b^{**}\| \leq \epsilon$ and every $f \in F$ and every $A \setminus S(b^{**}) \neq \emptyset$ we have that $\Phi(f_A(f(b))) > \Phi(b^{**})$.*

*Proof.* Fix $f \in F$ and let $\delta$ be as promised by the previous lemma, i.e. for every $\|z - b^{**}\| \leq \delta$ and every $A \setminus S(b^{**}) \neq \emptyset$ we have that $\Phi(f_A(z)) > \Phi(b^{**})$. Since $f(b^{**}) = b^{**}$ and $f$ is continuous there exists $\epsilon > 0$ so that $\|b - b^{**}\| \leq \epsilon$ implies $\|f(b) - f(b^{**})\| = \|f(b) - b^{**}\| \leq \delta$ and thus $\Phi(f_A(f(b))) > \Phi(b^{**})$. Now take the minimum $\epsilon$ over the finitely many $f \in F$. $\square$

**Definition.** *a sequence of sets $A_t \subseteq \mathcal{B}$ is called $T$-live if for every $i$ and for every $t$ there exists some $t \leq t^* \leq t + T$ such that $i \in S_{t^*}$.*

**Lemma 15.** *Fix a sequence $b = (b_t)$ where $b_{t+1} = f_{A_t}(b_t)$ such that the sequence $A_t$ is $T$-live. There are no condensation points of $(b_t)$ outside of $ME$.*

*Proof.* Assume that exists a condensation point $b^{**} \notin ME$ and a subsequence that converges to it, then $\Phi(b^{**}) < \Phi(b^*)$. Notice that $\Phi(b_t)$ is a non-decreasing sequence and so $\Phi(b_t) \leq \Phi(b^{**})$ for all $t$. Let $F$ be a set of functions achieved by composition of at most $T$ functions from $\{f_A | A \subset S(b^{**})\}$. So for every $f \in F$ we have that $f(b^{**}) = b^{**}$, while for every $B \setminus S(b^{**}) \neq \emptyset$ we have that $\Phi(f_B(b^{**})) > \Phi(b^{**})$. Let $\epsilon$ be as promised by the previous lemma, i.e., for every $\|b - b^{**}\| \leq \epsilon$ and every $f \in F$ and every $B$ such that $B \setminus S(b^{**}) \neq \emptyset$ we have that $\Phi(f_B(f(b))) > \Phi(b^{**})$. Since the subsequence converges to $b^{**}$ there exists $t_j$ in the subsequence so that $\|b_{t_j} - b^{**}\| \leq \epsilon$. Now let $t > t_j$ be the first time that $A_t \setminus S(b^{**}) \neq \emptyset$. Now $b_{t+1} = f_{A_t}(f(b_{t_j}))$, where $f$ is the composition of all $f_A$ for the times $t_j$ to $t$. We can now apply the previous lemma to get that $\Phi(b_{t+1}) = \Phi(f_{A_t}(f(b_{t_j})) > \Phi(b^{**})$, a contradiction. $\square$

**Lemma 16.** *Fix a sequence $b = (b_t)$ where $b_{t+1} = f_{A_t}(b_t)$ such that the sequence $A_t$ is $T$-live. Then, it holds that $\lim_{t \to \infty} d(b_t, ME) = 0$.*

*Proof.* By lemma 1 we have that $\Phi(b^{t+1}) \geq \Phi(b^t)$ making the sequence $\Phi(b^t)$ monotone and bounded from above ($\Phi(\cdot)$ is a bounded function). Hence it converges to some limit $L$. Either $L = \Phi^*$ or $L < \Phi^*$. In the former case, the result is immediate by lemma 12 and $b^t$ approaches the set $ME$. We show that the latter yields a contradiction. If $L < \Phi^*$, this implies that $b^t$ is bounded away from $ME$, i.e. there exists $\epsilon_0 > 0$ such that for all $t$ $d(b^t, ME) \geq \epsilon_0$. To see why this is true for all $t$ and not just in the limit, we observe that since the sequence $\Phi(b^t)$ is monotone, if we have $d(b^T, ME) = 0$ at some time $T$, then we have $\Phi(b^t) = \Phi^*$ for all $t > T$, which we currently assume is not the case. Therefore, every subsequence of $b^t$ is bounded away from $ME$, implying that every condensation point of $b^t$ is not in $ME$. The sequence $b^t$ is bounded and therefore has a converging subsequence with a condensation point not in $ME$, which is a contradiction to the previous lemma. $\square$

Regarding convergence of prices, as stated in section 2, equilibrium prices for each Fisher market are unique and attained by any bid profile $b^* \in ME$, thus, since the prices are a continuous function of the bids, the convergence of bids to this set implies the convergence of prices $p^t \to p^*$.

The last lemma concludes our proof of the Theorem 1.

$\square$

## Appendix E    Generic Markets

*Proof.* (Theorem 5): Assume by way of contradiction that a generic linear Fisher market has two distinct market equilibrium bid profiles $b^* \neq b^{**}$. For any market equilibrium $b$ it must hold that: (1) $\forall j \quad \sum_i b_{ij} = p_j^*$ since equilibrium prices are unique, and (2) $\forall i \quad \sum_j b_{ij} = B_i$ by budget feasibility. As $b^* \neq b^{**}$, there exists a pair $(i, j)$ with $b_{ij}^* \neq b_{ij}^{**}$, meaning that buyer $i$ has a different bid on good $j$ between $b^*$ and $b^{**}$, and so by (1) it must be that exists a buyer $k$ whose bid on good $j$ was also changed so that the price $p_j^*$ remains fixed; formally, $b_{kj}^* \neq b_{kj}^{**}$. In such case, by (2) there must be a good $\ell$ for which buyer $k$ has a different bid as well, since it's budget $B_k$ is fixed and fully utilized; formally $b_{k\ell}^* \neq b_{k\ell}^{**}$. As the graph $\Gamma = \{\mathcal{B} \cup \mathcal{G}, E\}$ with $E = \{\{i, j\} | b_{ij}' \neq b_{ij}^*\}$ is finite, following the process described above while satisfying the constraints (1) and (2) must lead to a cycle in the graph $\Gamma$.

Finally, we will show that there exists a market equilibrium with a cycle in its corresponding graph. Define $b' = \lambda b^* + (1 - \lambda)b^{**}$ for some $\lambda \in (0, 1)$ and note that $b'$ is also market equilibrium as the set of market equilibria is a convex set (see the model section in the main text). Let $\Gamma(b') = \{\mathcal{B} \cup \mathcal{G}, E(b')\}$ with $E(b') = \{\{i, j\} | b_{ij}' > 0\}$ be the corresponding graph of $b'$. Observe that $E \subseteq E(b')$ since if $b_{ij}^* \neq b_{ij}^{**}$ then it must be that $b_{ij}^* > 0$ or $b_{ij}^{**} > 0$ and in any such case $b_{ij}' > 0$. Thus, the graph $\Gamma(b')$ contains a cycle, contradicting Lemma 2 from the main text $\square$

*Proof.* (Lemma 2): Assume for the sake of contradiction that exists a cycle $C$ in $\Gamma(b^*)$, w.l.o.g. name the vertices of buyers and goods participating in the cycle in an ascending order; that is, $C = b_1 g_1 b_2 g_2 \ldots b_{k-1} g_k b_1$, where $b_i$ and $g_i$ represent buyers and goods $i$, respectively. Recall that for any market equilibrium if $x_{ij}^* > 0$ then $\frac{a_{ij}}{p_j^*} = c_i$ for some constant $c_i$ (see the model section in the main text). Applying this to the cycle $C$ yields the following equations. (1) By considering edges from buyers to goods $b_i \to g_i$ we obtain for $i \in [k-1] \quad a_{i,i} = c_i p_i^*$, and (2) by considering edges from goods to buyers $g_i \to b_{i+1}$ we obtain for $i \in [k-1] \quad a_{i+1,i} = c_{i+1}p_i^*$ and the edge closing the cycle yields $a_{1,k} = c_1 p_k^*$. Finally, by considering the product of ratios between valuations of buyers participating in the cycle we have the following condition.

$$
\frac{a_{21}}{a_{11}} \frac{a_{32}}{a_{22}} \frac{a_{43}}{a_{33}} \cdots \frac{a_{i+1,i}}{a_{i,i}} \cdots \frac{a_{k,k-1}}{a_{k-1,k-1}} \frac{a_{1,k}}{a_{k,k}} = \frac{c_2 p_1^*}{c_1 p_1^*} \frac{c_3 p_2^*}{c_2 p_2^*} \frac{c_4 p_3^*}{c_3 p_3^*} \cdots \frac{c_{i+1} p_i^*}{c_i p_i^*} \cdots \frac{c_k p_{k-1}^*}{c_{k-1} p_{k-1}^*} \frac{c_1 p_k^*}{c_k p_k^*}
$$
$$
= \frac{c_2}{c_1} \frac{c_3}{c_2} \frac{c_4}{c_3} \cdots \frac{c_{i+1}}{c_i} \cdots \frac{c_k}{c_{k-1}} \frac{c_1}{c_k}
$$
$$
= 1,
$$

which contradicts the genericity condition. $\square$

