# OpenReview forum: "Asynchronous Proportional Response Dynamics: Convergence in Markets with Adversarial Scheduling"
_NeurIPS.cc/2023/Conference — NeurIPS 2023 poster_

### Official Review · Reviewer_pfur · 2023-07-03

**Soundness:** 4 excellent
**Presentation:** 4 excellent
**Contribution:** 3 good
**Rating:** 8
**Confidence:** 4

**Summary:**

This paper studies asynchronous proportional response dynamics (PRD) in linear Fisher markets under adversarial scheduling. The authors proposed an associated game with specific player utilities which admits an exact potential function. Then, the authors show that the set of pure NE of the associated game is the same as the set of market equilibrium bids, which is also the same as the set of maximizers of the potential function. Next, the authors show that the best response dynamics (BRD), where a single player is activated in each round, converges to the equilibrium prices. In terms of PRD with subsets of active players in each round, the authors discussed its connection with BRD and showed that PRD strictly increases the potential function value unless there is no update in bids. Finally, they show that a “generic” linear Fisher market (i.e., no multiplicative equality/degeneracy) exhibits unique equilibrium bids. With the above developments, and through mathematical analysis arguments, the authors prove the main theorem: if the market is generic, and if players are activated in subsets (arbitrarily, but each at least once in every $T$ rounds), then PRD converges to the unique market equilibrium.

**Strengths:**

- The authors presented many interesting results that will likely be key foundational results for future research in game and market dynamics.
- These results are presented clearly with highly informative proof sketches and clear connections to other results.

**Weaknesses:**

None that I could think of. See **Questions**.

**Questions:**

For a non-generic market, does async PRD lead to price convergence, that is, $p^t = \sum_i b^t_{ij} \rightarrow p^*$? Can this be stated as a corollary somewhere in this work? Either way, it would be good to point this out (immediate, require some derivation, or still unclear).

Editorial suggestions.

- Line 236: $\Phi$ undefined (yet)?
- Line 329: Extra “a”.
- [nit] I would use "limit point" instead pf "condensation point" when discussing the convergence of bids, as the latter is slightly uncommon, although I believe it's used in Rudin's textbook.
- Appendix Line 316: “works” should be "work"? The authors are discussing the implications of multiple previous lemmas.

**Limitations:**

N.A.

More details:
- This work do not have (hidden, unstated) limitations. All claims in the abstract have been addressed in the work.
- This work is on theoretical properties of (variants of) well-konwn market dynamics and do not have immediate or potential negative societal impacts.

---

> ### Author Rebuttal · Authors · 2023-08-03
>
> Thank you for your feedback. Regarding the question about price convergence in non-generic markets: We believe that you are correct and prices do converge in general for PRD (as we did show for best-reply dynamics). However, we were not able to prove this with our current techniques and so the status is still unclear. We will point this out in the paper.
>
>
> Editing comments: Thank you for these helpful comments, we will incorporate them into the paper.

---

> > ### Comment · Reviewer_pfur · 2023-08-18
> >
> > Great, thanks for the response!

---

### Official Review · Reviewer_j9g9 · 2023-07-04

**Soundness:** 3 good
**Presentation:** 2 fair
**Contribution:** 2 fair
**Rating:** 4
**Confidence:** 2

**Summary:**

In this paper, the authors examined Proportional Response Dynamics (PRD) in linear Fisher markets in a setting where participants act asynchronously. In particular, they considered a setting where at each step, an adversary selects a subset of players to update their bids. The paper showed that in the generic case, if each bidder individually uses the PRD update rule when included in the selected group, then the entire dynamic converges to the market equilibrium. As part of their proof, they have also established other properties such as uniqueness of the market equilibrium and the convergence of best-response dynamics in an associated game.

**Strengths:**

1. This paper studies an interesting setup of the linear Fisher market where the activation can be asynchronous.
2. The theoretical results of this paper appear quite sound. The authors adopted novel proof techniques; in particular, they established an important connection between the associated game and the original game, which helped prove the convergence of asynchronous PRD.


**Weaknesses:**

1. One thing that appears missing from the current paper is the motivation behind studying the linear Fisher market with asynchronous PRD. It is essential to provide a rationale for studying this particular setting and justify why it is important. Since the authors only allow an intermediate level of allowed asynchrony, it becomes even more crucial to justify why this specific setting is worth investigating.
2. The organization of the paper is unclear, especially regarding the relationship between the results presented in Sections 3-5 and the proof of Theorem 1. I find it a bit difficult to follow the flow and understand which results contribute to proving Theorem 1 and which are significant on their own. Additionally, the role of Section 4 is unclear to me in the overall discussion of the paper (it seems to me that this section investigate some property of the associated game; but I don't see how it contributes to other results).
3. The definition of the "generic case" needs more clarity. It would be helpful to provide more explanation as to why this assumption is necessary and what it signifies in the context of the paper.
4. The authors mentioned that the convergence of PRD under the full asynchrony model remains unclear. It would be beneficial to specify the main challenges associated with achieving convergence in this model. Additionally, it'd be good if the authors could elaborate on their conjecture that convergence occurs if information delays are bounded.
5. If the convergence of PRD is shown by the potential function of the associated game being strictly increasing, is it true that the convergence could be arbitrarily slow? This might be related to the second open question (i.e., no speed of convergence result is provided).


**Questions:**

See weaknesses.

**Limitations:**

See weaknesses.

---

> ### Author Rebuttal · Authors · 2023-08-03
>
> Thank you for your feedback. We address below the specific points raised in the review.
>
>
> Motivation: Natural dynamics in markets (such as the PRD that we study) can be viewed as the aggregate emergent outcome of joint simple learning strategies of the participants. We believe that studying the emergent outcome of multiple interacting learning agents is an important and timely research area in machine learning, both theoretically in general and for economic applications. We do not suggest that we have a concrete application for this particular model but rather that we are advancing the theoretical study of such dynamics which are of significant general interest.
>
>
> Organization of the paper: Thank you for this comment. We will try to improve and clarify the organization of the paper further.
>
>
> Generic markets: The genericity condition is described in the introduction and formally defined in Definition 3. Intuitively, this means that there are no degeneracies in the market parameters (such as multiple values that are exactly equal), and we show that the absence of such degeneracies implies the uniqueness of the equilibrium. This in turn allows showing convergence to a point. Additionally, as mentioned in the conclusion, if there is some randomness in the process generating these parameters, the market would be generic with high probability. We will add further clarification in the introduction and include a reference to the formal definition.
>
>
> Main open problem: A significant challenge in the full asynchrony model is that one cannot directly employ potential-function arguments, as it may no longer be true that the standard potential function improves in every step. The motivation behind the conjecture that convergence still occurs when information delays are bounded is that the bids update in small steps. Thus, there is hope that over an epoch (a series of updates where everyone has updated their bids at least once), the potential will improve. However, we have not analyzed this model in this paper and leave such an analysis for future work.
>
>
> Speed of convergence: As we discuss in the paper, our current techniques allowed us to prove convergence with asynchronous bid updates but not the speed of convergence. We actually suspect that convergence is in polynomial time (in the problem parameters and 1/epsilon), but such an analysis may require some new ideas and we view it as an interesting and natural direction for further work.

---

### Official Review · Reviewer_pHCS · 2023-07-06

**Soundness:** 4 excellent
**Presentation:** 3 good
**Contribution:** 2 fair
**Rating:** 6
**Confidence:** 4

**Summary:**

The paper studies the convergence of Proportional Response Dynamics in linear Fisher Markets.
Fisher Markets are markets consisting of m divisible goods that should be shared among n agents with a linear utility function on items. The market not only must to decide the allocation, but it must also assign a payment to each agent for the fraction of each good she receives. We want that all items are sold (market clearing), that no agent spends more than her own budget (budget feasibility), and we are willing to allocate items so that there is no alternative market clearing and budget feasible allocation that an agent would prefer to the returned allocation (equilibrium).

It is known that a market equilibrium exists and can be computed in polynomial time by a centralized algorithm. Moreover there is a decentralized algorithm (known as tatonnement dynamics) that enable agents to quickly converge to the equilibrium, even if an adversary can choose at each time step which non-empty subset of agents would apply the dynamics update rule (subject to some fairness constraint).

In this work instead it is considered another dynamics (proportional response dynamics) that has been proved to converge when all agents update at each time step, but it was unknown whether this convergence property holds even against the above described adversary. This paper addresses this problem, by providing a positive answer.

The result is proved by observing that PRD for linear Fisher markets are equivalent to best response dynamics for a suitable potential game.

**Strengths:**

The paper is well written and presentation is very well-articulated and clear.
While the result re-uses some ideas previously established, it built an original framework to prove the main theorems on top of these ideas.

**Weaknesses:**

My first doubt about this paper is the relevance of this paper for this venue. While Fisher markets have been a very successful topic in economical and game theoretic literature, and computational aspects related to these markets have been of interest for theoretical computer science, these markets have not attracted very much the attention of AI community, maybe for the scarce practical applications (this can be seen also from the reference list of the paper). The paper does not make any effort to justify the study of these market within this community, neither it provides relevant applications.

My second doubt is about the relevance of these results. Why should we interested in proportional response dynamics if there is already a distributed dynamics that is known to converge to equilibrium even in an adversarial setting, that does it quickly, and that is a quite natural dynamics? Is there some motivation behind PRD that is not common also to tatonnement? The paper cites about similarity to a learning approach. Why this cannot be said also about tatonnement? And why should we interested to PRD convergence without bound on convergence time whenever we know that another dynamics converges quickly?

Based on this last comment it would be interesting to see in the experimental session a comparison about convergence of PRD and convergence of tatonnement, that is instead absent.

**Questions:**

See above.

**Limitations:**

Partially

---

> ### Author Rebuttal · Authors · 2023-08-03
>
> Thank you for your feedback. We address below the specific points raised in the review.
>
>
> Relevance: While we agree of course that our analysis is theoretical, we believe that the paper is relevant to NeurIPS: Natural dynamics in markets (such as the PRD that we study) can be viewed as the aggregate emergent outcome of joint simple learning strategies of the participants.  We believe that studying the emergent outcome of multiple interacting learning agents is an important and timely research area in machine learning, both theoretically in general and for economic applications.  Additionally, we note that the NeurIPS call for papers this year explicitly includes "Algorithmic Game Theory" in its list of topics.
>
>
> Motivation for exploring PRD: We agree, there is no doubt that tatonnement may also be viewed as a type of multi-agent learning, although with "goods players" responding to the excess demand rather than bidders responding to prices. Both models are interesting and have been widely studied. We believe that there is still much work left to be done on both dynamics, especially considering the perspective mentioned above of emergent behavior by multiple interacting learning agents.

---

> > ### Comment · Reviewer_pHCS · 2023-08-14
> >
> > Thank you for your answer.
> >
> > I need a further clarification on the second point: Do you believe there is a reason (either psychological or computational, or whatever else) for which PRD makes more sense than tatonnement in Fisher markets?

---

> > > ### Author Response · Authors · 2023-08-16
> > >
> > > Thank you for your comment. PRD and tatonnement describe different market behaviors, and so both are of interest as two alternative models. In particular, it has been argued in the literature that PRD has features of efficiency and simplicity that make the PRD model particularly interesting for the Fisher market model: Unlike tatonnement, in PRD, all goods are cleared at every step of the dynamic.  Secondly, the buyers in PRD do not need to solve an optimization problem at every step to determine their strategy in the next step. Instead, they simply divide their budget proportionally based on their last observed gains, and they do not need any parameters requiring tuning to do so.
> > >
> > > As we mention (e.g., in the introduction and related literature section), both tatonnement and PRD are considered as important dynamics in Fisher markets, and both have been broadly studied, and specifically, the question that we analyze of asynchronous PRD has been noted by several authors as an open problem. We believe our analysis provides novel and relevant theoretical results on this open problem, and it also raises directions for further work, which we discuss. Following your question, we will try to clarify further the differences between tatonnement and PRD in the paper to make this motivation clearer.

---

> > > > ### Comment · Reviewer_pHCS · 2023-08-16
> > > >
> > > > Thank you very much for clarification.
> > > >
> > > > I would suggest to include in your introduction motivations for studying PRD (despite of positive results about tatonnement).
> > > >
> > > > Accordingly, I am going to increase my score about the paper.

---

### Official Review · Reviewer_h7DH · 2023-07-06

**Soundness:** 3 good
**Presentation:** 3 good
**Contribution:** 3 good
**Rating:** 7
**Confidence:** 1

**Summary:**

This paper studies the problem of convergence in Proportional Response Dynamics (PRD) in linear Fisher markets when participants update in the dynamics under adversarial scheduling, i.e. an adversary specifies which subset of agents update their dynamics in a given round, subject to the constraint that each agent must be activated once every $T$ rounds at least. By leveraging auxiliary games and potential functions, it is shown that these asynchronous dynamics converge in generic settings; moreover, this analysis and connections also show that other natural dynamics also converge.

**Strengths:**

This work makes substantial progress on an open problem on the convergence of adversarially scheduled PRD. En route to this result, this work also derives implications for other dynamics by exploiting new structural properties of auxiliary games they consider for the analysis. In general, this paper is quite well-written and the arguments in the main text are well-explained.

**Weaknesses:**

While this work makes substantial progress, it feels like attaining rates of convergence should be achievable under some adaptation of the (seemingly new) arguments provided here; however, the current analysis relies on compactness-style arguments so do not seem directly amenable to answering this question.

**Questions:**

---While I am not an expert on this particular setting, all in all, this paper seems to make significant progress on an open problem in convergence of PRD. Moreover, the analysis appears both conceptually clean while novel (or rather, a nice twist on an existing approach), which may lead to further progress towards the remaining open problems highlighted in this work. I would defer to other reviewers on the significance of these results.

---The related work section and discussion appeared quite extensive, though I (again) am not an expert on this particular line of work.

---I could not verify all the details in the Supplementary Material; however, the arguments sketched in the main text seemed to make reasonable sense.

---Towards making progress on the main open problem posed in this work (i.e. total asynchronicity with delays), are there intermediate versions of this that may be amenable to a similar analysis? For instance, if there are stochastic or deterministic (i.e. nonadversarial) delays?

---

> ### Author Rebuttal · Authors · 2023-08-03
>
> Thank you for your feedback. Regarding the question about intermediate asynchrony models, we agree that intermediate levels of asynchrony with information delays are an interesting avenue to explore which can be useful for making progress in (or at least gaining insights on) the analysis of full asynchrony. Considering stochastic delays is an interesting direction that we have not yet explored. Another approach to consider when examining intermediate models is to start the analysis with very limited adversarial information delays (e.g., delays of a single step). We believe that it might be possible to extend our methods to handle such adversarial delays, but we have not pursued this route in the present paper and leave this analysis for future work.

---

> > ### Comment · Reviewer_h7DH · 2023-08-18
> >
> > Thanks for your response (and sorry for the delay)! I have no further questions at this time and will leave my score as is for now.

---

### Official Review · Reviewer_zQMj · 2023-07-11

**Soundness:** 4 excellent
**Presentation:** 4 excellent
**Contribution:** 4 excellent
**Rating:** 7
**Confidence:** 4

**Summary:**

Summary: The paper studies the Fisher market model, where there is a set of m sellers and n buyers. Each seller brings a unit of a divisible commodity for sale and each buyer brings a budget. The vendors value the money while the buyers value the commodities (goods).  A substantial amount of research has been done to develop methods for computing market equilibria and to understand the computational complexity of this problem.

This paper makes progress on understanding a well known type of dynamics called proportional response dynamics, where the market starts in some initial configuration and evolves over time. The agents modify their bids based on past engagements with other agents and former market state(s). Specifically, in proportional response dynamics, each agent adjusts their bids relative to the utility value of the goods in  the preceding round. Previous studies demonstrated that proportional response dynamics gravitate towards market equilibria in diverse scenarios; these dynamics are also interpretable.

Synchronous dynamics, where the players adapt their strategies with the same speed were studied extensively in the past work, but the more realistic asynchronous setting is less well understood (and before this paper not understood at all for the family of proportional response dynamics). The contribution of the paper is to study asynchronous proportional response dynamics in a very general setting where at each step a subset of players, adversarially chosen, update their strategies, with the constraint that each player has some minimum frequency of responding.

The main results of the paper are:

(1) For generic linear Fisher markets, proportional response dynamics with adversarial activation asynchrony, where each player is activated at least once every T steps, converge to the unique market equilibrium.

(2) The paper also finds a connection between the Fisher market and an associated game, such that the set of market equilibria of the Fisher market are the same as the set of Nash equilibria of the associated game and, furthermore, the same with the set of points that maximize a potential function \Phi (for which synchronous proportional response was shown in prior work to behave as mirror descent on this function). The paper shows that every proportional response step by any subset of players increases the potential function \Phi.

Evaluation: The paper makes several nice and significant contributions in a fundamental setting, contributing to the development of a theory of markets with learning agents. Such dynamics capture settings such as the stock market, which evolve over time. The paper raises intriguing questions for further study, such as how far asynchrony can be pushed, quantifying the rate of convergence depending on the extent of asynchrony, and investigating the correspondence identified in the paper between the proportional response dynamics and the associated game.

Various comments:

Page 2, line 60: “intermdiate” level of asynchrony -> intermediate

Page 6, line 236: "\Phi is it’s potential" -> its potential

Page 7, Lemma 1: "for all" should be capitalized -> For all

General suggestion -- some long inequalities or inequalities with fractions of sums are inlined (see e.g. page 5 line 205 in the main text, or page 3 in the appendix line 92 or page 5 in the appendix) and they are harder to read for this reason, it would be nicer to use displaymath or another environment like that. There are also some long paragraphs (e.g. when introducing the Fisher market) which could be divided into several shorter paragraphs for improved readability.



**Strengths:**

+ The paper makes several nice and significant contributions in a fundamental setting.
+ The paper is clearly written and of interest to researchers working in the space of games and learning.
+ Raises good questions for future work, such as how far asynchrony can be pushed, quantifying the rate of convergence depending on the extent of asynchrony, and investigating the correspondence identified in the paper between the proportional response dynamics and the associated game.


**Weaknesses:**

- Rate of convergence is not shown.

**Questions:**

There is something that I don't understand and may be missing. Does the associated game have a compact strategy space? For instance if all players bid zero on a good, then the price of that good is zero. The best response is not well defined at such a strategy profile, as  there can always be an improved response by a player - for instance, if the price of a certain good is zero and the player finds the good appealing, the player could bid 1 to acquire all of the good, but then bidding 1/2 would still enable them to secure the entire good while leaving some budget to increase the bid on other goods, and so forth. Essentially, the game seems to exhibit discontinuities at the strategy profiles where the price of a certain good is zero. Is Jensen's theorem applied to this game? If so, how does it work? (specifically, how does condition (1) in Jensen paper play out).

Note there are theorems that deal with games with discontinuous payoffs (e.g. https://link.springer.com/article/10.1007/s00199-015-0934-3 and https://www.jstor.org/stable/41237788).


**Limitations:**

The authors have acknowledged the limitations of the paper and explicitly stated the open problems remaining to be solved, including analyzing the rate of convergence, studying even stronger degree of asynchrony, and so forth.

---

> ### Author Rebuttal · Authors · 2023-08-03
>
> Thank you for your feedback. Yes, the strategy space is compact, it forms the full polytope where each buyer allocates its budget arbitrarily among the items. While you are correct that the best-reply function with respect to the standard utility function is formally undefined at zero, what we need is for the best-reply function to the *associated* utility to be continuous, and this is indeed the case. We note that the best reply to the associated utility function does not imply minimizing the bid on a good for which all the others bid zero, but rather there is a finite bid that equalizes the bang-per-buck. We will add a clarification about this in Section 4 and emphasize this further in the proof of Theorem 4.
>
>
> Editing comments: Thank you for these helpful comments, we will incorporate them into the paper.

---

> > ### Comment · Reviewer_zQMj · 2023-08-16
> >
> > Thank you for the clarification, I don't have further questions.

---

### Author Rebuttal · Authors · 2023-08-03

We thank all the reviewers for their valuable feedback, we will use it to improve the paper. We reply to specific points of each reviewer separately.

---

### Decision · Program_Chairs · 2023-09-21

**Decision:**

Accept (poster)

**Comment:**

This paper examines the proportional response (PR) algorithm for Fisher markets in the presence of delays and asynchronicities. The authors' main result is that, in generic linear Fisher markets where each player is activated with positive frequency over time (but otherwise possibly adversarially), the PR dynamics converge to the unique market equilibrium of the game. The authors also examine the convergence of the best-response dynamics (with a single agent updating at each stage), and they provide a range of numerical experiments to validate their findings.

The reviewers appreciated the paper's technical contributions, and the authors addressed the reviewers' concerns satisfactorily during the discussion phase. Most of the discussion revolved around the motivation for using the PR dynamics over tatonnement processes (which, however, may cycle in linear markets) but, in the end, there were no objections to an "accept" recommendation. I concur with this assessment, and I am happy to recommend acceptance as well.

In addition to the reviewers' comments, I would like to draw the authors attention to the extensive literature on the convergence of mirror descent in asynchronous / delayed settings. The PR algorithm is a special case of mirror descent with entropic regularization (I believe this observation goes back at least to Birnbaum, Devanur and Xiao, EC 2011) and, as such, I would invite the authors to discuss possible connections with the work of Quanrud and Khashabi online learning with delays ("Online learning with adversarial delays", NIPS 2015) and Zhou et al. ("Countering feedback delays in multi-agent learning", NIPS 2017) and Héliou et al. ("Gradient-free online learning in continuous games with delayed rewards", ICML 2020) in the multi-agent case.